# Latent SDEs on Homogeneous Spaces

**Sebastian Zeng, Florian Graf, Roland Kwitt**
University of Salzburg, Austria
{sebastian.zeng, florian.graf, roland.kwitt}@plus.ac.at

## Abstract

We consider the problem of variational Bayesian inference in a latent variable model where a (possibly complex) observed stochastic process is governed by the solution of a latent stochastic differential equation (SDE). Motivated by the challenges that arise when trying to learn an (almost arbitrary) latent neural SDE from data, such as efficient gradient computation, we take a step back and study a specific subclass instead. In our case, the SDE evolves on a homogeneous latent space and is induced by stochastic dynamics of the corresponding (matrix) Lie group. In learning problems, SDEs on the unit $n$-sphere are arguably the most relevant incarnation of this setup. Notably, for variational inference, the sphere not only facilitates using a truly uninformative prior, but we also obtain a particularly simple and intuitive expression for the Kullback-Leibler divergence between the approximate posterior and prior process in the evidence lower bound. Experiments demonstrate that a latent SDE of the proposed type can be learned efficiently by means of an existing one-step geometric Euler-Maruyama scheme. Despite restricting ourselves to a less rich class of SDEs, we achieve competitive or even state-of-the-art results on various time series interpolation/classification problems.

## 1 Introduction

Learning models for the dynamic behavior of a system from a collection of recorded observation sequences over time is of great interest in science and engineering. Applications can be found, e.g., in finance [5], physics [46] or population dynamics [2]. While the traditional approach of adjusting a prescribed differential equation (with typically a few parameters) to an observed system has a long history, the advent of neural ordinary differential equations (ODEs) [10] has opened the possibility for learning (almost) arbitrary dynamics by parameterizing the vector field of an ODE by a neural network. This has not only led to multiple works, e.g., [16; 27; 37; 60], elucidating various aspects of this class of models, but also to extensions towards stochastic differential equations (SDEs) [26; 30; 34; 58; 59], where one parametrizes the *drift* and *diffusion* vector fields. Such *neural SDEs* can, by construction, account for potential stochasticity in a system and offer increased flexibility, but also come at the price of numerical instabilities, high computational/storage costs, or technical difficulties when backpropagating gradients through the typically more involved numerical integration schemes.

While several recent developments to address these challenges exist, such as [30] or [26], it is important to note that neural ODEs or SDEs are most often used in settings where they serve as a model for some sort of (unobserved) latent dynamics. Such *latent ODE* or *latent SDE* models, cf. [10; 30; 48], hinge on the premise that the available observations are determined by an underlying continuous-time latent state (that might be subject to uncertainty). If one postulates that the observed dynamics are indeed governed by *simpler* latent dynamics that are easier to model, the question naturally arises whether

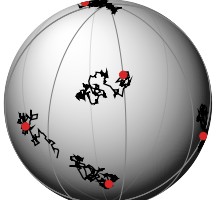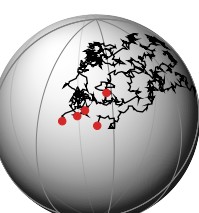

**Figure 1:** Sample paths from a prior (**left**) and posterior (**right**) process on the sphere. Initial values are marked by ●.

37th Conference on Neural Information Processing Systems (NeurIPS 2023).

a smaller subclass of SDEs, which can be solved efficiently and with less technical difficulty, may suffice for accurately modeling real-world phenomena. Specifically promising are SDEs that arise from the action of a Lie group on a homogeneous space, as they come with (1) an intuitive geometric interpretation, (2) a rich theory [1; 12; 22], and (3) easy-to-implement numerical solvers with strong convergence guarantees [35; 36; 40; 47].

**Contribution.** In this work, we take a step back from trying to learn (in theory) arbitrary SDEs and instead consider the very subclass mentioned above, i.e., SDEs that evolve on a homogeneous latent space as the result of the Lie group action. Numerical solutions to such SDEs are computed with a one-step geometric Euler-Maruyama scheme. The latter, combined with the assumption of a low latent space dimension enables backpropagating gradients through the SDE solver during learning via the "discretize-then-optimize" strategy. In the variational Bayesian learning regime, we instantiate our construction by learning latent SDEs on the unit $n$-sphere where the formula for the Kullback-Leibler (KL) divergence (within the evidence lower bound) between approximate posterior processes and an uninformative prior becomes surprisingly simple. Experiments on a variety of datasets provide empirical evidence that the subclass of considered SDEs is sufficiently expressive to achieve competitive or state-of-the-art performance on various tasks.

## 2 Related work

Our work is mainly related to advances in the neural ODE/SDE literature and recent developments in the area of numerical methods for solving SDEs on Lie groups.

**Neural ODEs and SDEs.** Since the introduction of neural ODEs in [10], many works have proposed extensions to the paradigm of parameterizing the vector fields of an ODE by neural networks, ranging from more expressive models [16] to higher-order variants [60]. As ODEs are determined by their initial condition, [27; 39] have also introduced a variant that can adjust trajectories from subsequent observations. Neural SDEs [19; 24; 34; 45; 54; 58; 59], meaning that an SDE's drift *and* diffusion coefficient are parametrized by neural networks, represent another natural extension of the neural ODE paradigm and can account for uncertainty in the distribution over paths. From a learning perspective, arguably the most common way of fitting neural ODEs/SDEs to data is the Bayesian learning paradigm, where learning is understood as posterior inference [29], although neural SDEs might as well be trained as GANs [26]. In the Bayesian setting, one may choose to model uncertainty (1) only over the initial latent state, leading to latent ODEs [10; 48], or (2) additionally over all chronologically subsequent latent states in time, leading to latent SDEs [19; 30]; in the first case, one only needs to select a suitable approximate posterior over the initial latent state whereas, in the second case, the approximate posterior is defined over paths (e.g., in earlier work [3] chosen as a Gaussian process). Overall, with recent progress along the line of (memory) efficient computation of gradients to reversible SDE solvers [25; 30], neural SDEs have become a powerful approach to model real-world phenomena under uncertainty.

**SDEs on matrix Lie groups.** In the setting of this work, SDEs are constructed via the action of a matrix Lie group on the corresponding homogeneous space. In particular, an SDE on the Lie group will translate into an SDE in the homogeneous space. Hence, numerical integration schemes that retain the Lie group structure are particularly relevant. Somewhat surprisingly, despite a vast amount of literature on numerical integration schemes for ODEs that evolve in Lie groups and which retain the Lie group structure under discretization, e.g., [21; 41; 42], similar schemes for SDEs have only appeared recently. One incarnation, which we will use later on, is the *geometric* Euler-Maruyama scheme from [35; 36] for Itô SDEs. For the latter, [47] established convergence in a strong sense of order 1. Stochastic Runge-Kutta–Munthe-Kaas schemes [40] of higher order were introduced just recently, however, at the cost of being numerically more involved and perhaps less appealing from a learning perspective. Finally, we remark that [44] introduces neural SDEs on Riemannian manifolds. However, due to the general nature of their construction, it is uncertain whether this approach can be transferred to a Bayesian learning setting and whether the intricate optimization problem of minimizing a discrepancy between measures scales to larger, high-dimensional datasets.

## 3 Methodology

### 3.1 Preliminaries

In the problem setting of this work, we assume access to a repeated (and partially observed) record of the continuous state history of a random dynamical system in observable space over time. The collected information is available in the form of a dataset of $N$ multivariate time series $\mathbf{x}^1, \ldots, \mathbf{x}^N$. In search for a reasonable model that describes the data-generating system, we consider a continuous stochastic process $X : \Omega \times [0, T] \to \mathbb{R}^d$ where each data sample refers to a continuous path $\mathbf{x}^i : [0, T] \to \mathbb{R}^d$. This path is assumed to be part of a collection of i.i.d. realizations sampled from the path distribution of the process $X$. As common in practice, we do not have access to the *entire* paths $\mathbf{x}^i$, but only to observations $\mathbf{x}^i(t) \in \mathbb{R}^d$ at a finite number of time points $t = t_k$ that can vary between the $\mathbf{x}^i$ and from which we seek to learn $X$.

Fitting such a process to data can be framed as posterior inference in a Bayesian learning setting, where the data generating process is understood as a latent variable model of the form shown in the figure below: first, a latent path $\mathbf{z}^i$ is drawn from a parametric prior distribution $p_{\boldsymbol{\theta}^*}(\mathbf{z})$ with true parameters $\boldsymbol{\theta}^*$; second, an observation path $\mathbf{x}^i$ is drawn from the conditional distribution $p_{\boldsymbol{\theta}^*}(\mathbf{x}|\mathbf{z}^i)$.

Essentially, this is the problem setting of [29]; however, the latent variables $\mathbf{z}$, as well as the observations $\mathbf{x}^i$ are path-valued and the true posterior $p_{\boldsymbol{\theta}}(\mathbf{z}|\mathbf{x})$ is intractable in general. To efficiently perform variational inference (e.g., with respect to $\mathbf{z}$) and to fit the model to observations, we choose a tractable approximate posterior $q_{\boldsymbol{\phi}}(\mathbf{z}|\mathbf{x})$ from some family of parametric distributions, a suitable prior $p_{\boldsymbol{\theta}}(\mathbf{z})$, and seek to maximize the evidence lower bound (ELBO)

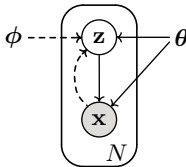

$$\log p_{\boldsymbol{\theta}}(\mathbf{x}^i) \geq \mathcal{L}(\boldsymbol{\theta}, \boldsymbol{\phi}; \mathbf{x}^i) = -D_{\mathrm{KL}}\left(q_{\boldsymbol{\phi}}(\mathbf{z}|\mathbf{x}^i) \| p_{\boldsymbol{\theta}}(\mathbf{z})\right) + \mathbb{E}_{\mathbf{z} \sim q_{\boldsymbol{\phi}}(\mathbf{z}|\mathbf{x}^i)}\left[\log p_{\boldsymbol{\theta}}(\mathbf{x}^i|\mathbf{z})\right] \quad . \quad (1)$$

This optimization problem is well understood in the context of vector-valued random variables, and can be efficiently implemented (provided the choice of prior and approximate posterior allows a *reparameterization trick*). However, the setting of a path-valued random latent variable comes with additional challenges.

**Parametric families of path distributions.** To efficiently perform approximate variational inference, we need to select a prior and approximate posterior in a reasonable and tractable manner from parametric families of distributions. To this end, we consider path distributions [11] of stochastic processes that are implicitly defined as strong solutions to linear Itô SDEs of the form

$$\mathrm{d}Z_t = f(Z_t, t)\,\mathrm{d}t + \sigma(Z_t, t)\,\mathrm{d}W_t, \quad t \in [0, T], \quad Z_0 \sim \mathcal{P} \quad . \quad (2)$$

$W$ is a vector-valued Wiener process, $f$ and $\sigma$ denote vector-valued drift and matrix-valued diffusion coefficients, respectively, and $Z_0$ is a multivariate random variable that follows some distribution $\mathcal{P}$. To ensure the existence of a unique, strong solution to Eq. (2), we require that $f, \sigma$ and $Z_0$ satisfy the assumptions of [57, Theorem 5.5]. As in the seminal work on neural ODEs [10], one may parameterize the drift and diffusion coefficients via neural networks, yielding *neural SDEs*. Given these prerequisites, maximizing the ELBO in Eq. (1) requires computing gradients with respect to $\boldsymbol{\theta}$ and $\boldsymbol{\phi}$, which hinges on sample paths from the approximate posterior process, generated by some (possibly adaptive step-size) SDE solver. Unfortunately, as mentioned in Section 1, modeling (2) with flexible drift and diffusion coefficients comes at high computational and memory cost. In the following (Section 3.2), we introduce our construction for SDEs on homogeneous spaces, discuss a simple SDE solver, and elaborate on the specific form of the KL divergence term in the ELBO of Eq. (1). In Section 3.3, we then instantiate the SDE construction on the example of the rotation group $\mathrm{SO}(n)$ acting on the unit $n$-sphere.

**Solving auxiliary tasks.** While learning SDEs of the form outlined above is an unsupervised problem, the overall approach can easily be extended to solve an auxiliary supervised task. For instance, we may have access to one target variable $y^i$ (e.g., a class-label) per time series or to one target variable per time point and time series, i.e., $y_{t_1}^i, \ldots, y_{t_m}^i$ (e.g., a class-label or some real-valued quantity per time point). In these scenarios, we will additionally learn to map latent states $\mathbf{z}$ to target variables, and extend the ELBO from Eq. (1) with an additive task-specific loss term. In practice, the overall optimization objective is then a weighted sum of (i) the KL divergence, (ii) the log-likelihood, and (iii) a task-specific supervised loss.

## 3.2 SDEs on homogeneous spaces

Drawing inspiration from manifold learning, e.g., [7; 14; 17], we assume the unobservable latent stochastic process $Z$ evolves on a low dimensional manifold $\mathbb{M} \subset \mathbb{R}^n$. If $\mathbb{M}$ admits a transitive action by a Lie group $\mathcal{G}$, i.e., if it is a homogeneous $\mathcal{G}$-space, then variational Bayesian inference can be performed over a family of processes that are induced by the group action. Specifically, we consider *one-point motions*, cf. [31], i.e., processes in $\mathbb{M}$ of the form $Z_t = G_t \cdot Z_0$ for some stochastic process $G$ in a matrix Lie group $\mathcal{G} \subset \mathrm{GL}(n)$ that acts by matrix-vector multiplication.

We define this stochastic process $G$ as solution to the linear SDE

$$\mathrm{d}G_t = \left( \mathbf{V}_0(t)\, \mathrm{d}t + \sum_{i=1}^{m} \mathrm{d}w_t^i \mathbf{V}_i \right) G_t, \quad G_0 = \mathbf{I}_\mathrm{n} \quad . \tag{3}$$

Herein, $w^i$ are independent scalar Wiener processes, $\mathbf{V}_1, \dots, \mathbf{V}_m \in \mathfrak{g}$ are fixed (matrix-valued) tangent vectors in the Lie algebra $\mathfrak{g}$, and $\mathbf{I}_\mathrm{n}$ is the $n \times n$ identity matrix. The drift function $\mathbf{V}_0 = \mathbf{K} + \mathbf{K}_\perp : [0, T] \rightarrow \mathbb{R}^{n \times n}$ consists of a tangential ($\mathfrak{g}$-valued) component $\mathbf{K}$ plus an orthogonal component $\mathbf{K}_\perp$ called pinning drift [36; 53]. It offsets the stochastic drift away from the Lie group $\mathcal{G}$ due to the quadratic variation of the Wiener processes $w^i$ and only depends on the noise components $\mathbf{V}_1, \dots, \mathbf{V}_m$ (and the geometry of $\mathcal{G}$). In case the Lie group $\mathcal{G}$ is quadratic, i.e., defined by some fixed matrix $\mathbf{P}$ via the equation $\mathbf{G}^\top \mathbf{P} \mathbf{G} = \mathbf{P}$, then the pinning drift $\mathbf{K}_\perp$ can be deduced directly from the condition $\mathrm{d}(\mathbf{G}^\top \mathbf{P} \mathbf{G}) = 0$ which implies that

$$\mathbf{V}_0(t) = \mathbf{K}(t) + \frac{1}{2} \sum_{i=1}^{m} \mathbf{V}_i^2 \quad , \qquad \text{and} \qquad \mathbf{V}_i^\top \mathbf{P} + \mathbf{P} \mathbf{V}_i = \mathbf{0} \quad . \tag{4}$$

We include the main arguments of the derivation of Eq. (4) in Appendix E.2 and refer to [6] and [12] for broader details.

Finally, given an initial value $Z_0 \sim \mathcal{P}$ drawn from some distribution $\mathcal{P}$ on the manifold $\mathbb{M}$, the SDE for $G$ induces an SDE for the one-point motion $Z = G \cdot Z_0$ via $\mathrm{d}Z_t = \mathrm{d}G_t \cdot Z_0$. Therefore, $Z$ solves

$$\mathrm{d}Z_t = \left( \mathbf{V}_0(t)\, \mathrm{d}t + \sum_{i=1}^{m} \mathrm{d}w_t^i \mathbf{V}_i \right) Z_t, \quad Z_0 \sim \mathcal{P} \quad . \tag{5}$$

**Numerical solutions.** To sample paths from a process $Z$ defined by an SDE as in Eq. (5), we numerically solve the SDE by a one-step geometric Euler-Maruyama scheme [36] (listed in Appendix E). Importantly, this numerical integration scheme ensures that the approximate numerical solutions reside in the Lie group, has convergence of strong order 1 [47], and can be implemented in a few lines of code (see Appendix D).

The idea of the scheme is to iteratively solve the SDE in intervals $[t_j, t_{j+1}]$ of length $\Delta t$, using the relation

$$Z_t = G_t Z_0 = (G_t G_{t_j}^{-1}) Z_{t_j} \quad . \tag{6}$$

Notably, $G_t G_{t_j}^{-1}$ follows the same SDE as $G_t$, see Eq. (3), but with initial value $\mathbf{I}_\mathrm{n}$ at $t = t_j$ instead of at $t = 0$. Moreover, if $\Delta t$ is sufficiently small, then $G_t G_{t_j}^{-1}$ stays close to the identity and reparametrizing $G_t G_{t_j}^{-1} = \exp(\Omega_t)$ by the matrix (or Lie group) exponential is feasible. The resulting SDE in the Lie algebra is *linear* and can thus be solved efficiently with an Euler-Maruyama scheme. Specifically, $\Omega_{t_{j+1}} = \mathbf{V}_0(t_j)\Delta t + \sum_{i=1}^{m} \Delta w_j^i \mathbf{V}_i$, where $\Delta w_j^i \sim \mathcal{N}(0, \Delta t)$ and we can differentiate through the numeric solver by leveraging the well-known reparametrization trick for the Gaussian distribution.

**Kullback-Leibler (KL) divergence.** For approximate inference in the variational Bayesian learning setting of Section 3.1, we seek to maximize the ELBO in Eq. (1). In our specific setting, the prior $p_{\boldsymbol{\theta}}(\mathbf{z})$ and approximate posterior distribution $q_{\boldsymbol{\phi}}(\mathbf{z}|\mathbf{x})$ of *latent paths* $\mathbf{z}$ are determined by an SDE of the form as in Eq. (5). In case these SDEs have the same diffusion term, i.e., the same $\mathbf{V}_i$, $i > 0$, then the KL divergence $D_{\mathrm{KL}}\left(q_{\boldsymbol{\phi}}(\mathbf{z}|\mathbf{x}) \| p_{\boldsymbol{\theta}}(\mathbf{z})\right)$ in the ELBO is finite and can be computed via Girsanov's theorem.

Specifically, if $\mathbf{V}_0^{\text{prior}}$ and $\mathbf{V}_0^{\text{post}}$ denote the respective drift functions and $\mathbf{V}_0^{\Delta} = \mathbf{V}_0^{\text{prior}} - \mathbf{V}_0^{\text{post}}$, then

$$D_{\text{KL}}\left(q_{\boldsymbol{\phi}}(\mathbf{z}|\mathbf{x}) \| p_{\boldsymbol{\theta}}(\mathbf{z})\right) = \frac{1}{2} \int_0^T \int_{\mathbb{M}} q_{Z_t}(\mathbf{z}) \left[\mathbf{V}_0^{\Delta}(t)\mathbf{z}\right]^{\top} \mathbf{\Sigma}^+(\mathbf{z}) \left[\mathbf{V}_0^{\Delta}(t)\mathbf{z}\right] \mathrm{d}\mathbf{z}\, \mathrm{d}t \ , \qquad (7)$$

where $q_{Z_t}$ is the marginal distribution of the approximate posterior process $Z$ at time $t$ and $\mathbf{\Sigma}^+(\mathbf{z})$ is the pseudo inverse of $\mathbf{\Sigma}(\mathbf{z}) = \sum_{i=1}^m \mathbf{V}_i \mathbf{z}\mathbf{z}^{\top}\mathbf{V}_i^{\top}$. For a derivation of this result, we refer to [43] and Appendix E.5.

### 3.3 SDEs on the unit $n$-sphere

The unit $n$-sphere $\mathbb{S}^{n-1}$ frequently occurs in many learning settings, e.g., see [13; 15; 32; 33; 38; 56]. It is a homogeneous space in context of the group $\mathrm{SO}(n)$ of rotations. For modeling time series data, $\mathbb{S}^{n-1}$ is appealing due to its intrinsic connection to trigonometric functions and, in a Bayesian learning setting, it is appealing since compactness allows selecting the uniform distribution $\mathcal{U}_{\mathbb{S}^{n-1}}$ as an uninformative prior on the initial SDE state $Z_0$.

The unit $n$-sphere also perfectly fits into the above framework, as $\mathrm{SO}(n)$ is the identity component of $\mathrm{O}(n)$, i.e., the quadratic Lie group defined by $\mathbf{P} = \mathbf{I}_n$. Its Lie algebra $\mathfrak{so}(n) = \left\{\mathbf{A} : \mathbf{A} + \mathbf{A}^{\top} = \mathbf{0}\right\}$ consists of skew-symmetric $n \times n$ matrices, and has a basis $\{\mathbf{E}_{kl} : 1 \leq k < l \leq n\}$ with $\mathbf{E}_{kl} = \mathbf{e}_k \mathbf{e}_l^{\top} - \mathbf{e}_l \mathbf{e}_k^{\top}$, where $\mathbf{e}_k$ is the $k$-th standard basis vector of $\mathbb{R}^n$. Importantly, if

$$\alpha^{\boldsymbol{\theta}} \neq 0 \ , \quad \{\mathbf{V}_i\}_{i=1}^{n(n-1)/2} = \left\{\alpha^{\boldsymbol{\theta}}\mathbf{E}_{kl}\right\}_{1 \leq k < l \leq n} \quad \text{and} \quad \mathbf{V}_0(t) = -(\alpha^{\boldsymbol{\theta}})^2 \frac{n-1}{2} \mathbf{I}_n \ ,$$

then Eq. (4) is fulfilled and the SDE in Eq. (5) becomes, after some transformations,

$$\mathrm{d}Z_t = -(\alpha^{\boldsymbol{\theta}})^2 \frac{n-1}{2} Z_t\, \mathrm{d}t + \alpha^{\boldsymbol{\theta}}(\mathbf{I}_n - Z_t Z_t^{\top})\, \mathrm{d}W_t \ , \quad Z_0 \sim p_{\boldsymbol{\theta}}(Z_0) = \mathcal{U}_{\mathbb{S}^{n-1}} \ , \qquad (8)$$

whose solution is a rotation symmetric and driftless[1] stochastic process on the unit $n$-sphere, i.e., the perfect *uninformative prior process* in our setting.

In fact, in case of $\alpha^{\boldsymbol{\theta}} = 1$, Eq. (8) is known as the *Stroock* equation [23; 55] for a spherical Brownian motion.

Similarly, we define an approximate posterior process as solution to an SDE with *learnable* drift and *learnable* (parametric) distribution $\mathcal{P}_0^{\boldsymbol{\phi}}$ on the initial state $Z_0$, i.e.,

$$\mathrm{d}Z_t^{\boldsymbol{\phi}} = \left(\mathbf{K}^{\boldsymbol{\phi}}(\mathbf{x})(t) - (\alpha^{\boldsymbol{\phi}})^2 \frac{n-1}{2} \mathbf{I}_n\right) Z_t\, \mathrm{d}t + \alpha^{\boldsymbol{\phi}}(\mathbf{I}_n - Z_t Z_t^{\top})\, \mathrm{d}W_t \ , \quad Z_0 \sim \mathcal{P}_0^{\boldsymbol{\phi}}(Z_0|\mathbf{x}) \ . \qquad (9)$$

Exemplary sample paths from solutions of Eq. (8) and Eq. (9) with constant drift are shown in Figure 1. We want to highlight that, in contrast to the parametric function $\mathbf{K}^{\boldsymbol{\phi}}(\mathbf{x}) : [0, T] \to \mathfrak{so}(n)$ (as part of the drift coefficient) and the parametric distribution $\mathcal{P}_0^{\boldsymbol{\phi}}$, the diffusion term does not depend on the evidence $\mathbf{x}$ but *only* on $\phi$. This allows selecting the same diffusion components $\alpha^{\boldsymbol{\phi}} = \alpha^{\boldsymbol{\theta}}$ in the posterior and prior process to ensure a finite KL divergence.

The latter has a particularly simple form (cf. Appendix E.5)

$$D_{\text{KL}}\left(q_{\boldsymbol{\phi}}(\mathbf{z}|\mathbf{x}) \| p_{\boldsymbol{\theta}}(\mathbf{z})\right) = D_{\text{KL}}\left(\mathcal{P}_0^{\boldsymbol{\phi}}(Z_0|\mathbf{x}) \| \mathcal{U}_{\mathbb{S}^{n-1}}\right) + \frac{1}{2(\alpha^{\boldsymbol{\theta}})^2} \int_0^T \int_{\mathbb{S}^{n-1}} q_{Z_t}(\mathbf{z}) \left\|\mathbf{K}^{\boldsymbol{\phi}}(\mathbf{x})(t)\mathbf{z}\right\|^2 \mathrm{d}\mathbf{z}\, \mathrm{d}t \ . \tag{10}$$

As customary in practice, we may replace the second summand with a Monte Carlo estimate and sum over all available time steps. Alternatively, using the upper bound

$$\int_{\mathbb{S}^{n-1}} q_{Z_t}(\mathbf{z}) \left\|\mathbf{K}^{\boldsymbol{\phi}}(\mathbf{x})(t)\mathbf{z}\right\|^2 \mathrm{d}\mathbf{z} \leq \left\|\mathbf{K}^{\boldsymbol{\phi}}(\mathbf{x})(t)\right\|_F$$

is equally possible. Notably, the second summand in the KL divergence of Eq. (10) has a quite intuitive interpretation: the squared norm is akin to a weight decay penalty, only that it penalizes a large (in norm) drift or, in other words, a group action that represents a large rotation.

---

[1]The drift only consists of the pinning component that is orthogonal to the Lie algebra $\mathfrak{so}(n)$.

### 3.4 Parameterizing the approximate posterior

As indicated in Eq.(9), $\phi$ enters the SDE associated with the approximate posterior $q_\phi(\mathbf{z}|\mathbf{x})$ in two ways: *first*, through the parametric distribution on initial states $Z_0$, and *second*, via the drift component $\mathbf{K}^\phi(\mathbf{x})$. For $\mathcal{P}_0^\phi$, we choose the power spherical [8] distribution $\mathcal{S}$ with location parameter $\boldsymbol{\mu}^\phi(\mathbf{x})$ and concentration parameter $\kappa^\phi(\mathbf{x})$, i.e., $\mathcal{P}_0^\phi(Z_0|\mathbf{x}) = \mathcal{S}(\boldsymbol{\mu}^\phi(\mathbf{x}), \kappa^\phi(\mathbf{x}))$. We realize $\boldsymbol{\mu}^\phi$, $\kappa^\phi$ and $\mathbf{K}^\phi$ by a task specific recognition network capable of learning a representation for time series $\mathbf{x}^i$ (with possibly missing time points and missing observations per time point).

In our implementation, we primarily use multi-time attention networks (mTAND) [50] to map a sequence of time-indexed observations to the parameters of the approximate posterior SDE. Vector-valued sequences are fed directly to mTAND, whereas image-valued sequences are first passed through a CNN feature extractor. Both variants are illustrated as types (A, ○) and (B, ○), resp., in Figure 2 and are used for interpolation tasks (i.e., imputation of missing observations at desired times) and in (per-time-point) classification problems. In situations where we need to extrapolate from just one *single* initial observation, we use type (C, ○) from Figure 2, which directly maps this image to a representation. In all cases, the so obtained representations are linearly mapped to the parameters $(\boldsymbol{\mu}^\phi, \kappa^\phi)$ of the power spherical distribution and to a skew-symmetric matrix $\mathbf{K}^\phi$ representing the drift at time 0 (and evolving in time as described below).

**Time evolution of the drift.** To ensure that $\mathbf{K}^\phi(\mathbf{x})(t)$ is continuous and can be evaluated for every $t \in [0, T]$, we use Chebyshev polynomials[2] as suggested by [37]. We found this choice suitable for our experiments in Section 4, but in principle any differentiable function $[0, T] \to \mathfrak{so}(n)$ can be used. In our implementation, a neural network maps a time series $\mathbf{x}$ to $K$ skew-symmetric matrices $\mathbf{K}_i^\phi(\mathbf{x}) \in \mathfrak{so}(n)$ and subsequently $\mathbf{K}^\phi(\mathbf{x})(t)$ is computed by

$$\mathbf{K}^\phi(\mathbf{x})(t) = \sum_{i=1}^{K} \mathbf{K}_i^\phi(\mathbf{x}) p_i(t) \ , \qquad (11)$$

where $p_i$ denotes the $i$-th Chebyshev polynomial.

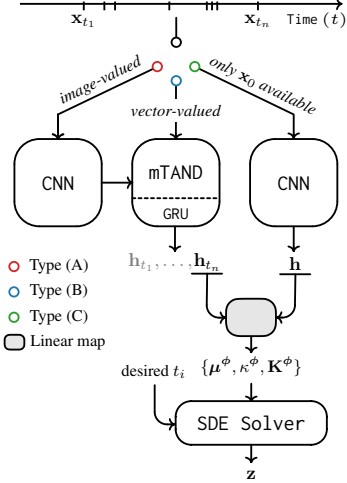

**Figure 2:** Recognition network types.

## 4 Empirical evaluation

We evaluate our latent SDE approach on different tasks from four benchmark datasets, ranging from time series interpolation to per-time-point classification and regression tasks. Full dataset descriptions are listed in Appendix A. While we stay as close as possible to the experimental setup of prior work for a fair comparison, we re-evaluated (using available reference implementations) selected results in one consistent regime, specifically in cases where we could not match the original results. In addition to the published performance measures, we mark the results of these re-runs in blue. For each method, we train *ten* different models and report the mean and standard deviation of the chosen performance measure. We note that if one were to report the same measures but over samples from the posterior, the standard deviations would usually be an order of magnitude smaller and can be controlled by the weighting of the KL divergence in the ELBO.

**Training & hyperparameters.** Depending on the particular (task-specific) objective, we optimize all model parameters using Adam [28] with a cyclic cosine learning rate schedule [18] (990 epochs with cycle length of 60 epochs and learning rate within [1e-6, 1e-3]). The weighting of the KL divergences in the ELBO is selected on validation splits (if available) or set to 1e-5 without any annealing schedule. We fix the dimensionality of the latent space to $n = 16$ and use $K = 6$ polynomials in Eq. (11), unless stated otherwise.

**Comparison to prior work.** We compare against several prior works that advocate for continuous-time models of latent variables. This includes the latent ODE approach of [10; 48] (ODE-ODE),

---

[2]Chebyshev polynomials are recursively defined via $p_{i+1}(t) = 2t p_i(t) - p_{i-1}(t)$ with $p_0(t) = 1$, $p_1(t) = t$.

the continuous recurrent units (CRU, f-CRU) work from [49], as well as the multi-time attention (mTAND-Enc, mTAND-Full) approach of [50]. On rotated MNIST, we additionally compare against the second-order latent ODE model from [60] and on the pendulum regression task, we also list the recent S5 approach from [52]. As [48] is conceptually closest to our work but differs in the implementation of the recognition network (i.e., using an ODE that maps a time series to the initial state of the latent ODE), we also assess the performance of a mTAND-ODE variant; i.e., we substitute the ODE encoder from [48] by the same mTAND encoder we use throughout. Finally, we remark that we only compare to approaches trainable on our hardware (see Appendix B) within 24 hours.

### 4.1  Per-time-point classification & regression

**Human Activity.**  This motion capture dataset[3] consists of time series collected from five individuals performing different activities. The time series are 12-dimensional, which corresponds to 3D coordinates captured by four motion sensors. Following the pre-processing regime of [48], we have 6,554 sequences available with 211 unique time points overall. The task is to assign each time point to one of seven (motion) categories. The dataset is split into 4,194 sequences for training, 1,311 for testing, and 1,049 for validation. We use a recognition model of type (B, ○) from Figure 2 and linearly map latent states to class predictions. For training, we minimize the cross-entropy loss evaluated on class predictions per available time point. Results are listed in Table 1.

Our primary focus in this experiment is to assess how a per-time-point classification loss affects our latent SDE model. As the linear classifier operates directly on the latent paths, the posterior SDEs need to account for possible label switches over time, e.g., a change of the motion class from *standing* to *walking*. Latent paths of sequences with only a *constant* class label across all time points must remain in a single decision region to minimize their error, while paths of sequences with label switches must cross decision boundaries and consequently increase their distance from the starting point; they literally must drift away.

**Table 1:** Human Activity.

|  | **Accuracy [%]** |
| --- | --- |
| [†]ODE-ODE [48] | $87.0 \pm 2.8$ |
| [†]mTAND-Enc [50] | $90.7 \pm 0.2$ |
| [†]mTAND-Full [50] | $\mathbf{91.1 \pm 0.2}$ |
| mTAND-ODE | $90.4 \pm 0.2$ |
| **Ours** | $90.6 \pm 0.4$ |

† indicates results from [50].

We visually assess this intuition after repeating the experiment with a decreased latent space dimension of $n = 3$ (i.e., $\mathbb{S}^2$), which reduces recognition accuracy only marginally to 90.2%.

A collection of *three* latent paths is shown in Figure 3. Each path is drawn from a posterior distribution computed from a different input time series at test time; the coloring visualizes the predicted class. Clearly visible, the latent paths corresponding to input samples with constant labels remain within the vicinity of their initial state (marked ● and ●, depending on the label). This suggests that the SDE is primarily driven by its diffusion term. Conversely, the latent path corresponding to an input sample with a label switch clearly shows the influence of the drift component. Our conjecture can be supported quantitatively by analyzing the path-wise KL divergences (which we measure relative to a *driftless* prior). Figure 3 shows the distributions of the latter over input samples *with* and *without* label switches. We see that latent paths corresponding to input samples *with* label switches exhibit higher KL divergences, i.e., they deviate more from the driftless prior.

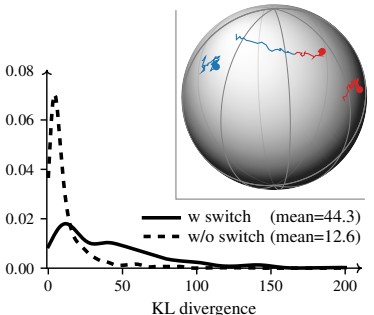

**Figure 3:** Latent space paths with and without label switches (on $\mathbb{S}^2$) and distribution of KL divergences.

**Pendulum regression.**  In this regression task [49], we are given (algorithmically generated [4]) time series of pendulum images at 50 irregularly spaced time points with the objective to predict the pendulum position, parameterized by the sine/cosine of the pendulum angle. The time series exhibit noise in the pixels and in the pendulum position in form of a deviation from the underlying physical model. We parameterize the recognition network by type (A, ○) from Figure 2 using the same CNN as in [49] and map latent states to (sine, cosine) tuples by the same decoder as in the latter work. 2,000 images are used for training, 1,000 images for testing and an additional 1,000 images for validation and hyperparameter selection.

---

[3]https://doi.org/10.24432/C57G8X

Table 2 lists the MSE computed on the (sine, cosine) encoding of the pendulum angle, averaged over all time points in the testing sequences. Notably, the recurrent architectures in Table 2, i.e., (f-)CRU and S5, can, by construction, condition on the data at observed time points and thus better "pin down" the temporal dynamic of the pendulum motion. Latent ODE/SDE models, on the other hand, need to integrate forward from one *initial* state and decode the angle from potentially erroneous latent states. In light of this observation, it is encouraging that the proposed latent SDE model performs only slightly worse than models based upon recurrent units (S5 and (f-)CRU). Nevertheless, Table 2 also reveals that performance on this task largely hinges on the network variant that encodes the initial latent state: in particular, although

**Table 2:** Pendulum *regression*.

|  | MSE $(\times 10^{-3})$ |
| --- | --- |
| [†]ODE-ODE [48] | $15.40 \pm 2.85$ |
| [†]mTAND-Enc [50] | $65.64 \pm 4.05$ |
| [†]CRU [49] | $4.63 \pm 1.07$ |
| [†]f-CRU [49] | $6.16 \pm 0.88$ |
| [‡]S5 [52] | $3.41 \pm 0.27$ |
| mTAND-ODE | $4.65 \pm 1.21$ |
| mTAND-Enc | $\mathbf{3.20 \pm 0.60}$ |
| CRU | $3.74 \pm 0.27$ |
| f-CRU | $\mathbf{3.25 \pm 0.26}$ |
| **Ours** | $3.84 \pm 0.35$ |

†, ‡ indicate results from [49] and [52], resp.

ODE-ODE [48] fails, switching the initial state encoding (to mTAND) yields substantially better results. Notably, our experiments[4] with mTAND-Enc [50] (which fails in the runs reported in [49]), yields results on a par with the state-of-the-art.

## 4.2 Interpolation

**PhysioNet (2012).** We consider the *interpolation task* of [50] with *subsampled* time points. The dataset [51] contains 8,000 time series of 41 patient variables, collected over a time span of 48 hours. Each variable is observed at irregularly spaced time points. In the considered experimental setup of [50], interpolation is understood as observing a fraction ($p$) of the available time points per sequence and interpolating observations at the remaining fraction $(1 - p)$ of time points. Importantly, this is more challenging than the setting in [49] or [48], as the experimental setup in the latter works corresponds to $p = 1$. Furthermore, the pre-processing (see Appendix A for details) of [50] renders results for $p = 1$ not *directly* comparable to the ones presented in [49]. As customary in the literature [20; 49; 50], we use 80% of all time series for training, 20% for testing, scale all variables to a range of $[0, 1]$, and map the full time span of 48 hours to the time interval $[0, 1]$ at a quantization level of either 1 [20; 48; 50] or 6 minutes [49]. As in the experiments of Section 4.1, the recognition model is parameterized by type (B, ○) from Figure 2 and latent states are decoded to observations as in [50].

Table 3 lists the MSE obtained on varying fractions ($p$) of time points available to the recognition model. In the situation of *no subsampling*, i.e., $p = 1$, the f-CRU and mTAND-Full models perform substantially better than ours. However, in that case, the task effectively boils down to reconstructing the observed input sequences. In contrast, *subsampling*, i.e., $p < 1$, truly allows assessing the *interpolation capability* on previously unseen data. In this situation, we achieve a new state-of-the art as our method improves upon its underlying mTAND module.

**Pendulum interpolation.** We evaluate on the same pendulum interpolation task as in [49]. This task differs from the position prediction task of Section 4.1 in that we observe only around half of 50 randomly chosen, but uncorrupted, time points per sequence. Interpolation is understood as predicting the pendulum images at the unseen time points. As remarked earlier, this is notably more challenging for latent ODE/SDE type approaches than for models that rely on recurrent units, as we cannot pin down the latent path at the observed measurements other than through the loss. In terms of recognition network ar-

**Table 4:** Pendulum *interpolation*.

|  | MSE $(\times 10^{-3})$ |
| --- | --- |
| [†]mTAND-Full [50] | $15.40 \pm 0.07$ |
| [†]ODE-ODE [48] | $15.06 \pm 0.14$ |
| [†]f-CRU [49] | $1.39 \pm 0.07$ |
| [†]CRU [49] | $\mathbf{1.00 \pm 0.07}$ |
| mTAND-ODE | $8.23 \pm 0.10$ |
| f-CRU | $2.61 \pm 0.19$ |
| CRU | $2.06 \pm 0.05$ |
| **Ours** | $8.15 \pm 0.06$ |

† indicates results from [49].

---

[4]We set the mTAND encoder such that its GRU returns an output for each point in time. These latent states are then mapped to an angle prediction by the same decoder as in [49].

**Table 3:** Results on the PhysioNet (2012) interpolation task from [50] for two quantization levels (**Top**: 6 min; **Bottom**: 1 min). Percentages indicate the fraction of *observed time points*. We report the MSE ($\times 10^{-3}$) on the observations at *unseen* time points, averaged across all testing sequences. Results for mTAND-ODE at 1 min quantization (bottom part) are *not* listed, as training time exceeded our 24-hour threshold.

| Observed % → | 50% | 60% | 70% | 80% | 90% | 100% |
|---|---|---|---|---|---|---|
| CRU [49] | $5.11 \pm 0.40$ | $4.81 \pm 0.07$ | $5.60 \pm 0.70$ | $5.54 \pm 0.61$ | $5.85 \pm 0.82$ | $1.77 \pm 0.92$ |
| f-CRU [49] | $5.24 \pm 0.49$ | $4.96 \pm 0.36$ | $4.85 \pm 0.10$ | $5.66 \pm 0.71$ | $6.11 \pm 0.82$ | $1.10 \pm 0.95$ |
| mTAND-Full [50] | $3.61 \pm 0.08$ | $3.53 \pm 0.06$ | $3.48 \pm 0.08$ | $3.55 \pm 0.12$ | $3.58 \pm 0.09$ | $\mathbf{0.49 \pm 0.03}$ |
| mTAND-ODE | $3.38 \pm 0.03$ | $3.33 \pm 0.02$ | $3.32 \pm 0.03$ | $3.33 \pm 0.04$ | $3.35 \pm 0.02$ | $1.82 \pm 0.04$ |
| **Ours** | $\mathbf{3.14 \pm 0.03}$ | $\mathbf{3.06 \pm 0.05}$ | $\mathbf{3.01 \pm 0.05}$ | $\mathbf{2.94 \pm 0.07}$ | $\mathbf{3.00 \pm 0.08}$ | $1.55 \pm 0.01$ |
| CRU [49] | $5.20 \pm 0.44$ | $5.25 \pm 0.55$ | $5.22 \pm 0.60$ | $5.23 \pm 0.59$ | $5.59 \pm 0.78$ | $1.82 \pm 0.97$ |
| f-CRU [49] | $5.53 \pm 0.54$ | $5.22 \pm 0.58$ | $5.41 \pm 0.65$ | $5.25 \pm 0.60$ | $5.80 \pm 0.76$ | $0.84 \pm 0.16$ |
| [†]mTAND-Full [50] | $4.19 \pm 0.03$ | $4.02 \pm 0.05$ | $4.16 \pm 0.05$ | $4.41 \pm 0.15$ | $4.80 \pm 0.04$ | $\mathbf{0.47 \pm 0.01}$ |
| **Ours** | $\mathbf{3.14 \pm 0.04}$ | $\mathbf{3.03 \pm 0.03}$ | $\mathbf{3.01 \pm 0.02}$ | $\mathbf{3.04 \pm 0.09}$ | $\mathbf{3.03 \pm 0.14}$ | $1.54 \pm 0.02$ |

† indicates results from [50] (which only lists the 1-min setting).

chitecture, we use type (A, ○) from Figure 2 and decode latent states via the convolutional decoder of [49]. Hyperparameters are selected based on the validation set.

As can be seen from Table 4, which lists the MSE across all images in the testing sequences, approaches based on recurrent units perform exceptionally well on this task, while the latent ODE approach, as well as mTAND-Full perform mediocre at best. Our approach yields decent results, but visual inspection of the reconstructed sequences (see visualization next to Table 4) shows slight phase differences, which underscores our remark on the difficulty of learning this fine-grained dynamic by integrating forward from $t = 0$ and guiding the learner only through the loss at the available time points.

**Rotating MNIST.** This dataset [9] consists of sequences of $28 \times 28$ grayscale images of gradually clockwise rotated versions of the handwritten digit "3". One complete rotation takes place over 16 evenly spaced time points. This gives a total sequence length of 16, where we interpret each partial rotation as a time step of $1/16$ in the time interval $[0, 1]$. During training, four images in each sequence are dropped randomly, and one rotation angle (i.e., the testing time point) is consistently left out. Upon receiving the first image at time $t = 0$, the task is to interpolate the image at the left-out time point $t = 3/16$.

**Table 5:** Rotating MNIST.

| | **MSE** $(\times 10^{-3})$ |
|---|---|
| [†]GPPVAE-dis | $30.9 \pm 0.02$ |
| [†]GPPVAE-joint | $28.8 \pm 0.05$ |
| [†]ODE$^2$VAE | $19.4 \pm 0.06$ |
| [†]ODE$^2$VAE-KL | $18.8 \pm 0.31$ |
| CNN-ODE | $14.5 \pm 0.73$ |
| **Ours** | $\mathbf{11.5 \pm 0.38}$ |

† indicates results from [60].

We follow the evaluation protocol of [60], where 360 images are used for training, 360 for testing, and 36 for validation and hyperparameter selection. The recognition model is of type (C, ○) from Figure 2 and uses the same convolutional encoder (CNN) as in [60], as well as the same convolutional decoder from latent states to prediction images. Table 5 reports the MSE at the testing time point, averaged across all testing sequences. Our model achieves the lowest MSE on this benchmark. Figure 4 additionally highlights the extrapolation quality of the model on the example of integrating the SDE learned for the interpolation task forward from $t = 0$ to $t = 4$ in steps of $1/16$.

Notably, this is the only experiment where we do not use $K = 6$ Chebyshev polynomials for the time evolution of the drift, see Eq. (11), but only $K = 1$, as we want to assess whether a constant velocity on the sphere is a suitable model for the constant rotation velocity in the data. Figure 4 confirms this intuition. In fact, using $K > 1$ polynomials yields better interpolation results but at a loss of extrapolation quality as the higher-order polynomials are less well-behaved for large $t$. Regarding the baselines, we do not compare against (f-)CRU and mTAND as they are not directly applicable here. As only the image at the *initial* time point is used as model input, learning any reasonable attention mechanism (mTAND) or conditioning on observations spread out over the observation time interval (CRU) is impossible.

**Runtime measurements.** For a detailed runtime analysis across different approaches that model an underlying (stochastic) differential equation, we refer the reader to Appendix C.1.

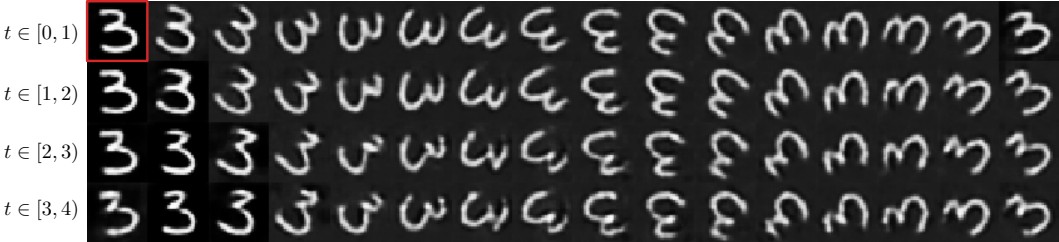

**Figure 4:** Exemplary reconstructions on the Rotating MNIST data. Shown are the results (on one testing sequence) by integrating forward from $t = 0$ (marked red) to $t = 4$, i.e., three times longer than what is observed ($t \in [0, 1)$) during training.

## 5 Discussion & Limitations

We have presented an approach to learning (neural) latent SDEs on homogeneous spaces with a Bayesian learning regime. Specifically, we focused on one incarnation of this construction, i.e., the unit $n$-sphere, due to its abundance in many learning problems. Our SDEs are, by design, less expressive than in earlier works. Yet, when modeling a latent variable, this limitation appears to be of minor importance (as shown in our experiments). We do, however, gain in terms of overall model complexity, as backpropagating through the SDE solver can be done efficiently (in a "discretize-then-optimize" manner) without having to resort to adjoint sensitivity methods or variable step-size solvers. While our approach could benefit from the latter, we leave this for future research. Overall, the class of neural SDEs considered in this work presents a viable alternative to existing neural SDE approaches that seek to learn (almost) arbitrary SDEs by fully parameterizing the vector fields of an SDEs drift and diffusion coefficient by neural networks. In our case, we do use neural networks as high-capacity function approximators, but in a setting where one prescribes the space on which the SDE evolves. Finally, whether it is reasonable to constrain the class of SDEs depends on the application type. In situations where one seeks to learn a model in observation space directly, approaches that allow more model flexibility may be preferable.

### Acknowledgments

This work was supported by the Austrian Science Fund (FWF) under project P31799-N38 and the Land Salzburg within the EXDIGIT project 20204-WISS/263/6-6022 and projects 0102-F1901166-KZP, 20204-WISS/225/197-2019.

> **Source code** is available at https://github.com/plus-rkwitt/LatentSDEonHS.

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

# Supplementary material

*In this supplementary material, we report full dataset descriptions, list architecture details, present additional experiments, and report any left-out derivations.*

## Contents

**Societal impact.**    This work mainly revolves around a novel type of neural SDE model. We envision no direct negative societal impact but remark that any application of such a model, e.g., in the context of highly sensitive personal data (such as ICU stays in PhysioNet (2012)), should be inspected carefully for potential biases. This is particularly relevant if missing measurements are interpolated and eventually used in support of treatment decisions. In this scenario, biases (most likely due to the training data) could disproportionately affect any minority group, for instance, as imputation results might be insufficiently accurate.

## A  Datasets

This section contains a description of each dataset we use in our experiments. While all datasets have been used extensively in the literature, reported results are often not directly comparable due to inconsistent preprocessing steps and/or subtle differences in terms of data partitioning as well as variable selection (e.g., for PhysioNet (2012)). Hence, we find it essential to highlight the exact data-loading and preprocessing implementations we use: for **Human Activity** and **PhysioNet (2012)** experiments, we rely on publicly available code from [50] (which is based upon the reference

implementation of [48]). For the **Pendulum** experiments, we rely on the reference implementation from [49], and for **Rotating MNIST**, we rely on the reference implementation of [60].

## A.1 Human Activity

This motion capture dataset[5], used in Section 4.1, consists of time series collected from five individuals performing different activities. The time series are 12-dimensional, corresponding to 3D coordinates captured by four motion sensors. Following the preprocessing regime of [48] (who group multiple motions into one of *seven* activities), we have 6,554 sequences available with 211 unique time points overall and a fixed sequence length of 50 time points. The time range of each sequence is mapped to the time interval $[0, 1]$, and no normalization of the coordinates is performed. The task is to assign each time point to one of the seven (motion/activity) categories (i.e., "lying", "lying down", "sitting", "sitting down", "standing up from lying", "standing up from sitting", "standing up from sitting on the ground"). The dataset is split into 4,194 sequences for training, 1,311 for testing, and 1,049 for validation. Notably, the majority of time series have *one* constant label, with only 239 (18.2%) of the time series in the testing portion of the dataset containing actual label switches (e.g., from "lying" to "standing up from lying").

## A.2 Pendulum

The pendulum dataset, used in Sections 4.1 and 4.2, contains algorithmically generated [4] (grayscale) pendulum images of size $24 \times 24$. For the **regression** task from [49], we are given time series of these pendulum images at 50 irregularly spaced time points with the objective to predict the pendulum position, parameterized by the (sine, cosine) of the pendulum angle. The time series exhibit noise in the pixels and in the pendulum position (in form of a deviation from the underlying physical model). For the **interpolation** task from [49], uncorrupted images are available, but we only observe a smaller fraction of the full time series (50% of time points randomly sampled per time series). The task is to interpolate images at the unobserved time points. Two exemplary time series (per task) from the training portion of the dataset are shown in Figure 5. For both tasks, we use 4,000 images for training, 1,000 images for testing and 1,000 images for validation.

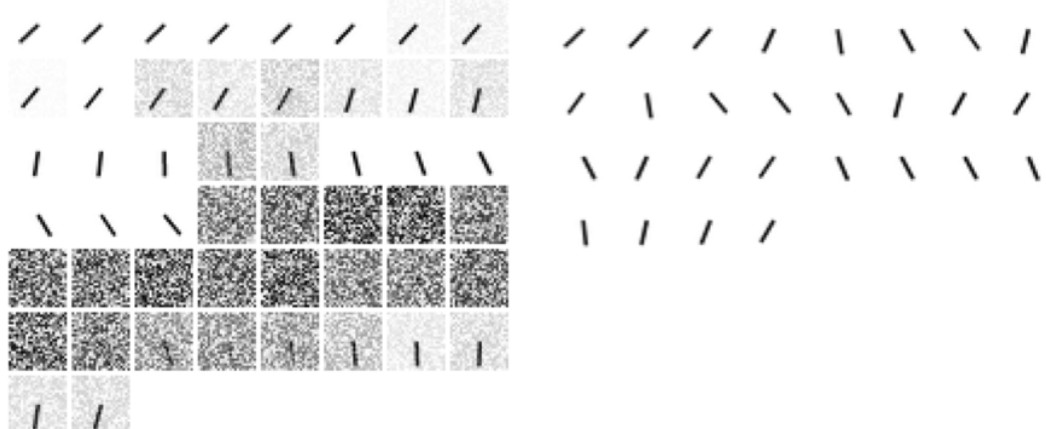

**Figure 5:** Illustration of *one* exemplary training time series from the (*left*) **regression** and (*right*) pendulum position **interpolation** task, respectively, from [49]. Note that the images shown here are inverted for better visualization.

## A.3 PhysioNet (2012)

The PhysioNet (2012) [51] dataset has been used many times in the literature. However, a clean comparison is difficult as the preprocessing steps, as well as the split and variable selection, often differ in subtle details. Hence, we focus on one evaluation setup [50] and re-evaluate prior work precisely in this setting instead of adopting the reported performance measures.

---

[5]https://doi.org/10.24432/C57G8X

The original challenge dataset contains time series for 12,000 ICU stays over a time span of 48 hours. 41 variables are available, out of which a small set is only measured at admission time (e.g., age, gender), and up to 37 variables are sparsely available within the 48-hour window. To the best of our knowledge, almost all prior works normalize each variable to $[0, 1]$ (e.g., [48; 49; 50] and many more). In [50], and hence in our work as well, *all* 41 variables are used, and only the training portion of the full challenge dataset is considered. The latter consists of 8,000 time series, split into 80% training sequences and 20% testing sequences. Depending on the experimental regime in prior works, observations are either aggregated within **1 min** [48; 50] or within **6 min** [49] time slots. This yields 480 and 2,880, resp., possible (unique) time points. Overall, the available measurements in this dataset are very sparse, i.e., only $\approx 2\%$ of all potential measurements are available.

While prior work typically focuses on *mortality prediction*, i.e., predicting one binary outcome per time series, we are interested in interpolating observations at unseen time points. To assess this *interpolation capability* of a model, we adopt the *interpolation task* of [50]. In this task, interpolation is understood as observing a fraction ($p$) of the (already few) available time points per sequence and interpolating observations at the remaining fraction $(1 - p)$ of time points. From our perspective, this is considerably more challenging than the setting in [49] or [48], as the experimental setup in the two latter works corresponds to $p = 1$.

Furthermore, the preprocessing regime in [50] slightly differs from [49] in two aspects: (1) in the way each variable is normalized and (2) in the number of variables used (41 in [50] and 37 in [49]). Regarding the normalization step: letting $x_1, \ldots, x_N$ denote all values of one variable in the training dataset, and $x_{\min}, x_{\max}$ denote the corresponding minimum and maximum values of that variable, [50] normalizes by $(x_i - x_{\min})/x_{\max}$, whereas [49] normalizes by $(x_i - x_{\min})/(x_{\max} - x_{\min})$. For fair comparison, we re-ran all experiments with the preprocessing of [50].

### A.4 Rotating MNIST

This dataset [9] consists of sequences of $28 \times 28$ grayscale images of gradually clockwise rotated versions of the handwritten digit "3". One complete rotation takes place over 16 evenly spaced time points. During training, one time point is consistently left-out (according to the reference implementation of [60], this is the 4th time point, i.e. at time $t = 3/16$) and four additional time points are dropped randomly per sequence. Upon receiving the first image at time $t = 0$, the task is to interpolate the image at the left-out (4th) time point. Exemplary training sequences are shown in Figure 6. We exactly follow the dataset splitting protocol of [60], where 360 images are used for training, 360 for testing and 36 for validation and hyperparameter selection.

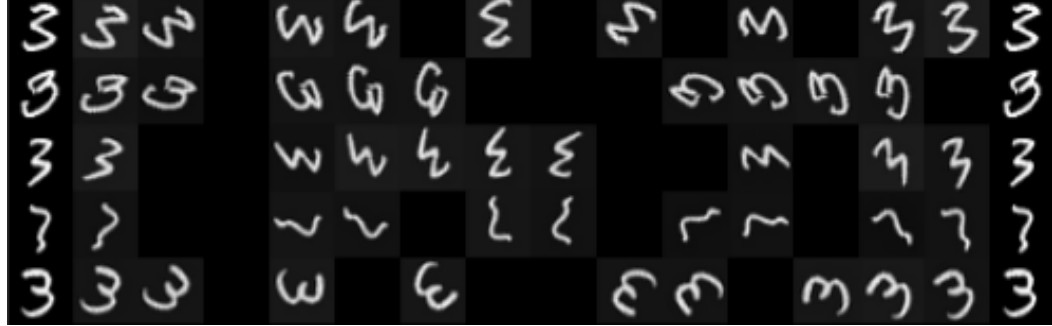

**Figure 6:** Five (counted in rows) exemplary training sequences from the rotating MNIST dataset.

## B  Architecture details

The three types of recognition networks (i.e., A, B, or C) we use throughout all experiments are shown in Figure 2. In the following, we specify the configuration of the networks (from Figure 2) per experiment, discuss the choice of loss function (in addition to the KL divergence terms from the ELBO), and provide details about the architecture(s) in works we compare to.

Irrespective of the particular type of network that processes an input sequence, our parameterization of the SDE on the $n$-sphere remains the same across all experiments. In particular, given a representation

of an input sequence, say $\mathbf{h}$, we linearly map to (1) the parameters of the power spherical distribution, i.e., the posterior distribution on the initial state, and (2) to the coefficients of $K$ Chebyshev polynomials, cf. Eq. (11), that control the time evolution of $\mathbf{K}^\phi$. Here, the superscript $\phi$ denotes the dependency of $\mathbf{K}$ on the (parameters of the) recognition network. As $\mathbf{K}^\phi$ is skew-symmetric and has, in case of $\mathrm{SO}(n)$, $n(n-1)/2$ free parameters, we linearly map to $Kn(n-1)/2$ coefficients overall. Unless noted otherwise, we consistently use $K = 6$ polynomials and set $n = 16$. In all experiments with ODE/SDE type models, we use ADAM [28] for optimization, with initial learning rate set to 1e-3 and subject to a cyclic cosine learning rate schedule (cycle length of 60 epochs) [18]. Finally, when replacing our SDE on the $n$-sphere by an ODE in $\mathbb{R}^n$ (denoted as mTAND-ODE), we use an Euler scheme as ODE solver and compute gradients via the adjoint sensitivity method (implemented in the publicly available torchdiffeq package).

## B.1   Human Activity

As recognition network we use type (B, ○) from Figure 2 with an mTAND encoder (mTAND-Enc [50]) configured as in the publicly-available reference implementation[6]. When used as a standalone module (i.e., without any subsequent ODE/SDE), mTAND-Enc uses the *outputs* of its final GRU unit (which yields representations in $\mathbb{R}^{128}$ at all desired time points) to linearly map to class predictions. In our setting, we take the *last hidden state* of this GRU unit as input $\mathbf{h} \in \mathbb{R}^{128}$ to the linear maps mentioned above. When replacing our SDE on the $n$-sphere by an ODE in $\mathbb{R}^n$ (denoted as mTAND-ODE), the last GRU state is linearly mapped to the location parameter of a Gaussian distribution and to the entries of a diagonal covariance matrix. For mTAND-ODE, the ODE is parameterized as in [48] by a 3-layer MLP (with ELU activations) with 32 hidden units, but the dimensionality of the latent space is increased to 16 (for a fair comparison). As in our approach, latent states are linearly mapped to class predictions. Notably, when comparing ODE-ODE [48] to mTAND-ODE, the only conceptual difference is in the choice of recognition network (i.e., an ODE in [48], *vs.* an mTAND encoder here).

**Loss components.**   In addition to the KL divergence terms in the ELBO, the *cross-entropy loss* is computed on class predictions at the available time points. For our approach, as well as mTAND-ODE, the selected weighting of the KL divergence terms is based on the validation set accuracy (and no KL annealing is performed).

## B.2   Pendulum regression & interpolation

In the *pendulum* experiments of Section 4.1 and Section 4.2, we parameterize the recognition network by type (A, ○) from Figure 2. Images are first fed through a convolutional neural network (CNN) with the same architecture as in [49] and the so obtained representations are then input to an mTAND encoder (mTAND-Enc). Different to the mTAND configuration in Appendix B.1, the final GRU unit is set to output representations $\mathbf{h} \in \mathbb{R}^{32}$. For the *regression task*, latent states are decoded via the same two-layer MLP from [49]. For *interpolation*, we also use the convolutional decoder from [49].

**Loss components.**   For the *regression task*, we compute the mean-squared-error (MSE) on the (sine, cosine) encodings of the pendulum angle *and* use a Gaussian log-likelihood on the image reconstructions. Although the images for the regression task are corrupted (by noise), we found that adding a reconstruction loss helped against overfitting. The weighting of the KL divergence terms in the ELBO is selected on the validation set (and no KL annealing is performed). For the *interpolation task*, the MSE term is obviously not present in the overall optimization objective.

**Remark on mTAND-Enc.**   While the mTAND-Enc approach from [50] fails in the experimental runs reported in [49] (see [†]mTAND-Enc in Table 2), we highlight that simply using the outputs of the final GRU unit as input to the two-layer MLP that predicts the (sine, cosine) tuples worked remarkably well in our experiments.

---

[6]https://github.com/reml-lab/mTAN

### B.3 PhysioNet (2012)

In the PhysioNet (2012) interpolation experiment, we parameterize our recognition network by type (B, ◯) from Figure 2 using representations $\mathbf{h} \in \mathbb{R}^3 2$ as input to the linear layers implementing the parameters of the drift and the initial distribution. The mTAND encoder is configured as suggested in the mTAND reference implementation for this experiment. We further adjusted the reference implementation[7] of (f-)CRU from [49] to use exactly the same data preprocessing pipeline as in [50] and configure (f-)CRU as suggested by the authors.

**Loss components.** In addition to the KL divergence terms, the ELBO includes the Gaussian log-likelihood of reconstructed observations at the available time points (using a fixed standard deviation of 0.01). The weightings of the KL divergence terms is fixed to 1e-5, *without* any further tuning on a separate validation portion of the dataset (and no KL annealing is performed).

### B.4 Rotating MNIST

In the rotating MNIST experiment, the recognition network is parameterized by type (C, ◯) from Figure 2. In particular, we rely on the convolutional encoder (and decoder) from [60], except that we only use the *first* image as input (whereas [60] also need the two subsequent frames for their momentum encoder). Unlike the other experiments, where the Chebyshev polynomials are evaluated on the time interval $[0, 1]$, we deliberately set this range to $[0, 20]$ in this experiment. This is motivated by the fact that we also seek to extrapolate beyond the time range observed during training, see Figure 4. Furthermore, we use only one polynomial ($K = 1$), which is reasonable due to the underlying "simple" dynamic of a constant-speed rotation; setting $K > 1$ only marginally changes the results listed in Table 5.

**Loss components.** Gaussian log-likelihood is used on image reconstructions and weights for the KL divergences are selected based on the validation set (and no KL annealing is performed).

### B.5 Computing infrastructure

All experiments were executed on systems running Ubuntu 22.04.2 LTS (kernel version `5.15.0-71-generic x86_64`) with 128 GB of main memory and equipped with either NVIDIA GeForce RTX 2080 Ti or GeForce RTX 3090 Ti GPUs. All experiments are implemented in PyTorch (v1.13 and also tested on v2.0). In this hardware/system configuration, all runs reported in the tables of the main manuscript terminated within a 24-hour time window.

## C   Additional empirical results

### C.1 Runtime measurements

We compare the runtime of our method to two close alternatives, where we substitute our latent (neural) SDE on the unit $n$-sphere, by either a latent (neural) ODE [10; 48] *or* a latent (neural) SDE [30] in $\mathbb{R}^n$. In other words, we exchange the model components for the latent dynamics in the approaches we use for Rotating MNIST and PhysioNet (2012), see Section 4. This provides insights into runtime behavior as a function of the integration steps, moving from 16 steps on Rotating MNIST to 480 on PhysioNet (2012) at a 6 min quantization level and to 2,880 steps on PhysioNet (2012) at a 1 min quantization level. In all experiments, we exclusively use Euler's method as ODE/SDE solver with a fixed step size. Results are summarized in Table 6. In particular, we list the elapsed real time for a *forward+backward* pass (batch size 50), averaged over 1,000 iterations.

---

[7] https://github.com/boschresearch/Continuous-Recurrent-Units

**Table 6:** Comparison of the elapsed real time of a *forward+backward* pass over a batch (of size 50) across different tasks/datasets (with varying number of integration steps). For each task, the overall model sizes (in terms of the number of parameters) are comparable. Listed values are averages over 1,000 iterations $\pm$ the corresponding standard deviations. We remark that the runtime of our approach when using $(\mathbb{R}^n, \mathrm{GL}(n))$ instead of $(\mathbb{S}^{n-1}, \mathrm{SO}(n))$ differs only marginally.

|  | Rotating MNIST | PhysioNet (2012) (6 min) | PhysioNet (2012) (1 min) |
|---|---|---|---|
| Number of integration steps | 16 | 480 | 2,880 |
| CNN/mTAND-ODE ([10; 48]) | $0.053 \pm 0.004$ | $0.926 \pm 0.017$ | $3.701 \pm 0.033$ |
| CNN/mTAND-SDE ([30]) | $0.112 \pm 0.009$ | $2.075 \pm 0.042$ | $10.273 \pm 0.173$ |
| **Ours** $-(\mathbb{S}^{n-1}, \mathrm{SO}(n))$ | $0.055 \pm 0.005$ | $0.179 \pm 0.008$ | $2.693 \pm 0.027$ |

Overall, although our approach implements a latent SDE, runtime per batch is on a par (or better) with the latent ODE variant on rotating MNIST and on PhysioNet (2012) at both quantization levels. Furthermore, we observe a substantially lower runtime of our approach compared to the (more flexible) latent SDE work of [30], which is closest in terms of modeling choice. It is, however, important to point out that the way our SDE evolves on the sphere (i.e., as a consequence of the group action) lends itself to a quite efficient implementation where large parts of the computation can be carried out as a single tensor operation (see Appendix D).

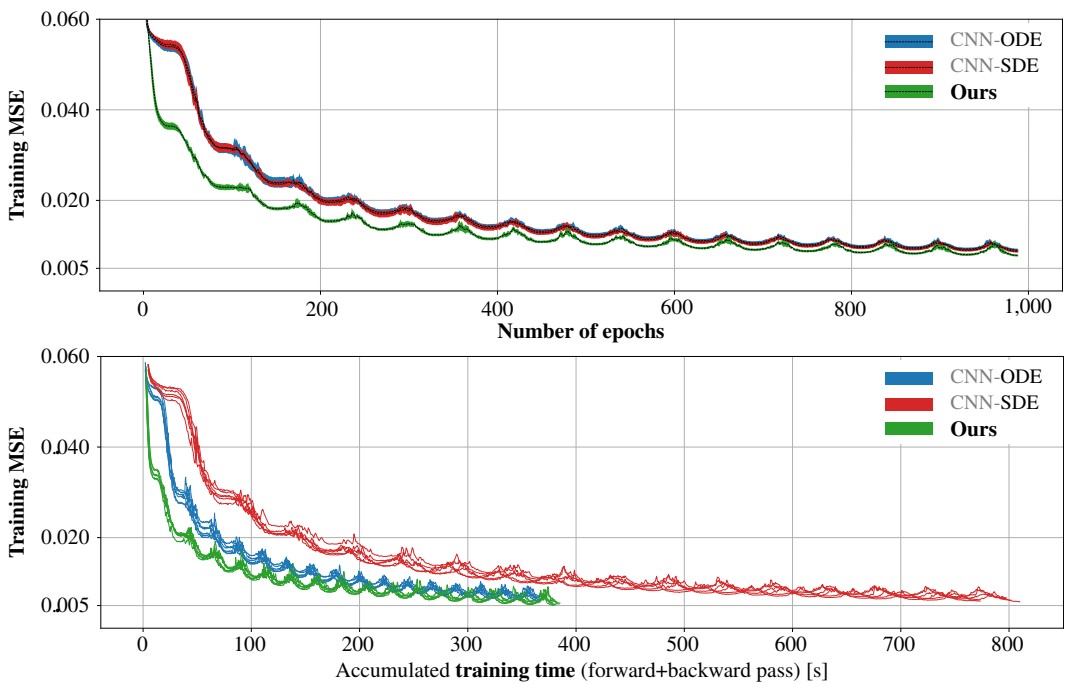

**Figure 7:** Training time comparison on Rotating MNIST for *five* different training runs per model. **Bottom:** Training MSE vs. (accumulated) training time of each model. **Top:** Training MSE (dashed) and standard deviations (shaded) *vs.* number of training epochs.

Aside from the direct runtime comparison above, we also point out that while a shorter (forward) runtime per batch ensures a faster inference time, it does not necessarily result in faster overall training time as the convergence speed might differ among methods. To account for the latter, we compare the evolution of training losses in Figure 7 when training on the Rotating MNIST dataset. The upper plot shows *training MSE vs. number of epochs* and reveals that our method needs fewer updates to converge, whereas the same model but with latent dynamics replaced by a latent ODE [10] or latent SDE [30] in $\mathbb{R}^n$ converge at approximately the same rate. The bottom plot accounts for

*different runtimes per batch* and reveals that our method needs only around half of the training time than the latent SDE [30] variant in $\mathbb{R}^n$.

We want to point out, though, that runtime is, to a large extent, implementation-dependent, and therefore, the presented results should be taken with caution. For the baselines, we used publicly available implementations of ODE/SDE solvers, i.e., https://github.com/rtqichen/torchdiffeq for the latent ODE [10] models and https://github.com/google-research/torchsde for the latent SDE [30] models.

**Comparison to non-ODE/SDE models.** Finally, we present a runtime comparison with methods that do not model an underlying (stochastic) differential equation, i.e., mTAND-Full and CRU. This relative runtime comparison was performed on the PhysioNet (2012) dataset and the runtimes/batch are averaged over batches of size 50; results are summarized in Table 7. The PhysioNet (2012) dataset is particularly interesting concerning runtime, as it contains the most time points (480 potential time points for a quantization level of 6 minutes and 2,880 time points for a quantization level of 1 minute).

**Table 7:** Relative timing results (*forward+backward* pass) on **PhysioNet (2012)** for both quantization levels (reported in the main manuscript).

|            | 6 min | 1 min |
|------------|-------|-------|
| CRU        | 22×   | 25×   |
| f-CRU      | 17×   | 20×   |
| mTAND-Full | 1×    | 1×    |
| **Ours**   | 5×    | 55×   |

*All runtime experiments were performed on the same hardware (see Appendix B.5 with a GeForce RTX 2080 Ti GPU).*

## C.2 Stochastic Lorenz attractor

To illustrate that our model is expressive enough to model difficult dynamics, we replicate the experiment by [30] of fitting a stochastic Lorenz attractor. Figure 8 shows 75 posterior sample paths.

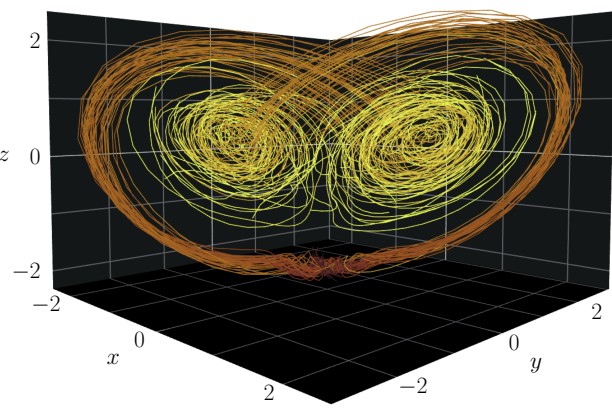

**Figure 8:** Shown are 75 posterior sample paths from our model on the stochastic Lorenz attractor data of [30].

## C.3 Uncertainty quantification

In the main part of this manuscript, we primarily focussed on the interpolation and regression/classification capabilities of our method on existing benchmark data. It is, however, worth pointing out that our generative model can be utilized for *uncertainty quantification*. We can sample from the probabilistic model outlined in Section 3 and use these samples to estimate the underlying distribution. While we did not evaluate the uncertainty quantification aspect as thoroughly as the regression/classification performance, we present preliminary qualitative results on the Pendulum angle regression experiment in Figure 9. This figure shows density estimates for the angle predictions on testing data. The spread of the angle predictions is (as one would expect) sensitive to the KL-divergence weight in the loss function. The present model was trained with a weight of 1e-3.

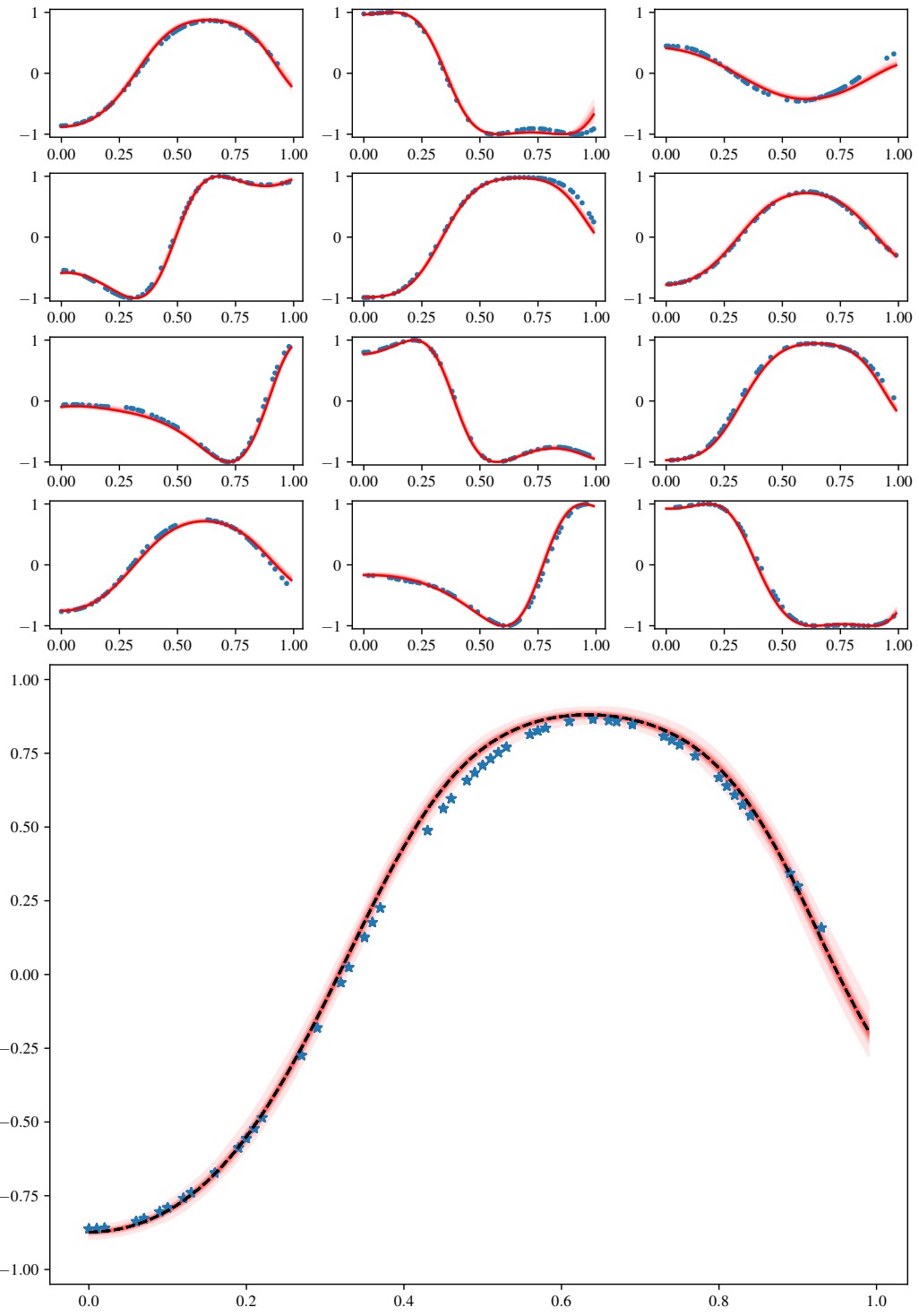

**Figure 9: Top**: Visualization of quality of fit and uncertainty for the first 12 testing time series in the pendulum regression task. Shown is the ground truth (in blue) and the predicted angle (sine). Shaded regions correspond to densities, estimated from quantiles of 1,000 draws from the posterior distribution. **Bottom**: Zoomed-in plot of the first time series.

## D  Implementation

To highlight that our approach can be implemented in a few lines of code, Listings 1, 2 and 3 provide a *minimal working example* (in PyTorch) of how to obtain solutions to an SDE on $\mathbb{S}^{n-1}$. The only part left out in this example is the generation of the time evolution of $\mathbf{K}$. To execute the code snippet in Listing 4, the reference implementation of the power spherical distribution[8] from [8] as well as the einops package[9] are needed.

---

**Listing 1** SDE solver in Lie algebra $\mathfrak{so}(n)$.

```
1    from einops import repeat
2
3    class PathSampler(nn.Module):
4        def __init__(self, dim):
5            super(PathSampler, self).__init__()
6            self.register_buffer("skew_sym_basis", gen_basis(dim))
7            self.alpha = nn.Parameter(torch.tensor(1./dim, dtype=torch.float32))
8
9        def forward(self, Kt, delta):
10           bs, nt = Kt.shape[0], Kt.shape[1]
11           Et = repeat(self.skew_sym_basis, 'k a b -> bs nt k a b', bs=bs, nt=nt)
12           noise = torch.randn(bs, nt, Et.size(2), 1, 1, device=Kt.device)
13           Vt = (noise*Et*delta.sqrt()*self.alpha).sum(2)
14           return Kt*delta + Vt
```

---

**Listing 2** Generate basis in $\mathfrak{so}(n)$.

```
1    def gen_basis(n):
2        r = int(n*(n-1)/2)
3        B = torch.zeros(r, n, n)
4        for nth, (i,j) in enumerate(torch.combinations(torch.arange(n))):
5            B[nth,i,j] = 1.
6        return B - B.transpose(1,2)
```

---

**Listing 3** Generate SDE solution on $\mathbb{S}^{n-1}$.

```
1    from torch.distributions import Distribution
2    from torch import Tensor
3
4    def sample(t: Tensor, pzx: Distribution, Kt: Tensor, sampler: PathSampler):
5        z0 = pzx.rsample().unsqueeze(-1)
6        delta = torch.diff(t)[0]
7        Ot = sampler(Kt, delta)
8        Qt = torch.matrix_exp(Ot.contiguous())
9
10       zi = [z0]
11       for t_idx in range(0, len(t)-1):
12           Qtz = torch.matmul(Qt[:,t_idx,:,:], zi[-1])
13           zi.append(Qtz)
14       return torch.stack(zi, dim=1).squeeze()
```

---

[8] https://github.com/nicola-decao/power_spherical
[9] https://einops.rocks/

**Listing 4** Example usage with dummy $\mathbf{K}, \boldsymbol{\mu}, \kappa$.

```python
import torch
import torch.nn as nn
from power_spherical import PowerSpherical

n = 16
batch_size = 4
num_time_points = 100

device = 'cuda' if torch.cuda.is_available() else 'cpu'
t = torch.linspace(0, 1, num_time_points, device=device)
sampler = PathSampler(n).to(device)

mu = torch.randn(batch_size, n, device=device)       # batch of μ's
kappa = torch.rand(batch_size, 1, device=device)     # batch of κ's
pzx = PowerSpherical(
    torch.nn.functional.normalize(mu),
    kappa.squeeze())

# Kt just contains random skew-symmetric matrices in this example
Kt = torch.randn(batch_size, len(t), n, n, device=device)
Kt = Kt - Kt.transpose(2,3)
zis = sample(t, pzx, Kt, sampler)
# zis is now a tensor of size (bs,num_time_points,n)
```

# E   Theory

In this part, we provide further details on the construction of SDEs on homogeneous spaces of quadratic matrix Lie groups $\mathcal{G}$, and derive the formula for the KL divergence between solutions of such SDEs that appeared in the main manuscript.

## E.1   Background on stochastic differential equations (SDEs)

We first make some initial remarks on stochastic processes and on stochastic differential equations (SDEs) in $\mathbb{R}^n$ and introduce the concepts and notations that are used throughout this final chapter of the supplementary material.

Given some probability space $(\Omega, \mathcal{F}, P)$, a stochastic process on $[0, T]$ with state space $\mathbb{R}^n$ is a function $X : \Omega \times [0, T] \to \mathbb{R}^n$ such that for each point of time $t \in [0, T]$ the map $X_t = X(\cdot, t) : \Omega \to \mathbb{R}^n$, $\omega \mapsto X(\omega, t)$ is measurable. In other words, $\{X_t : t \in [0, T]\}$ is a collection of random variables $X_t$ indexed by $t$, cf. [66, Chapter I.1; 57, Chapter 2.1; 63, Chapter 1.1]. Conversely, each sample $\omega \in \Omega$ defines a function $X(\omega, \cdot) : [0, T] \to \mathbb{R}^n$, $t \mapsto X(\omega, t)$ called a realization or sample path. Thus, a stochastic process models randomness in the space of functions $[0, T] \to \mathbb{R}^n$, or in a "reasonable" subspace thereof. In more formal terms, the law or path distribution $\mathbb{P}^X$ associated to a stochastic process $X$ is defined as the push-forward of the measure $P$ to this function space [66, Chapter I.3].

Arguably the most important stochastic process is the Wiener process $W$ which formalizes the physical concept of the Brownian motion, i.e., the random motion of a small particle inside a fluid caused by thermal fluctuations, see [68, Chapter 3] for a heuristic approach to the topic. For stochastic processes $X$ that are sufficiently well-behaved, one can define a stochastic integral $\int_0^t X_s \, \mathrm{d}W_s$, called Itô integral of $X$ with respect to $W$, cf. [66, Chapter IV.2; 57, Chapter 3.1; 63, Chapter 3.2]. Notably, the result, is again a stochastic process so that one can implicitly define stochastic processes via an Itô integral equation of the form

$$X_t = X_0 + \int_0^t f(X_s, s) \, \mathrm{d}s + \int_0^t \sigma(X_s, s) \, \mathrm{d}W_s \tag{12}$$

often denoted as an Itô stochastic differential equation (SDE)

$$\mathrm{d}X_t = f(X_t, t) \, \mathrm{d}t + \sigma(X_t, t) \, \mathrm{d}W_t, \quad X_0 = \xi \ . \tag{13}$$

Here, $f : \mathbb{R}^n \times [0, T] \to \mathbb{R}^n$ is called the drift coefficient, $\sigma : \mathbb{R}^n \times [0, T] \to \mathbb{R}^{n \times m}$ is called the diffusion coefficient, and the initial value $\xi$ is a random variable that is independent of the Wiener process $W$. Further information about different notions of solutions to an SDE, i.e., strong and weak solutions, as well as a comprehensive consideration of their existence and uniqueness can be found in [66, Chapter 9; 57, Chapter 5; 63, Chapter 5]. Additionally, for a treatment of SDEs from a numerical perspective, we recommend [64].

In this work, we consider time series data as realizations of a stochastic process, i.e., as samples from a function-valued random variable as described in Section 3.1. To perform variational inference in such a setting, we utilize SDEs to construct function-valued distributions. Specifically, we consider non-autonomous linear Itô SDEs with multiplicative noise that evolve on a homogeneous space of a (quadratic) matrix Lie group. To ensure the existence and uniqueness of strong solutions, the drift and diffusion coefficients that appear in the considered SDEs satisfy the Lipschitz and growth conditions of [57, Theorem 5.2.1]

**A remark on notation.**   As outlined in Section 3.4, we aim to learn the parameters $\phi$ of an approximate posterior. Thus, the SDEs we consider are not specified by an *initial random variable* $X_0 = \xi$, but by an *initial distribution* $\mathcal{P}^{X_0}$ such that $X_0 \sim \mathcal{P}^{X_0}$. Therefore, we utilize the notation

$$\mathrm{d}X_t = f(X_t, t) \, \mathrm{d}t + \sigma(X_t, t) \, \mathrm{d}W_t \ , \quad X_0 \sim \mathcal{P}^{X_0} \ . \tag{14}$$

A (strong) solution to such an SDE is then understood as a stochastic process $X$ that *(i)* is adapted (see [66, Definition 4.2]), *(ii)* has continuous sample paths $t \mapsto X(\omega, \cdot)$, *(iii)* almost surely satisfies the integral equation in Eq. (12), and *(iv)* there is a random variable $\xi \sim \mathcal{P}^{X_0}$ that is independent of the Wiener process $W$ such that $X_0 = \xi$ holds almost surely. In accordance with [57, Theorem 5.2.1], we demand that and the initial distribution $\xi$ has finite second moment, but, of course, distinct choices of $X_0 = \xi$ might define distinct strong solutions $X$.

## E.2 SDEs in quadratic matrix Lie groups

**Definition 1** (cf. [65]). *Let $\mathbb{R}^{n\times n}$ be the vector space of real $n \times n$ matrices. We denote the general linear group, i.e., the group of invertible real $n \times n$ matrices equipped with the matrix product, as*

$$\mathrm{GL}(n) = \left\{ \mathbf{A} \in \mathbb{R}^{n\times n} \mid \det \mathbf{A} \neq 0 \right\} . \tag{15}$$

$\mathrm{GL}(n)$ *is a Lie group with Lie algebra*

$$\mathfrak{gl}(n) = \left( \mathbb{R}^{n\times n}, [\cdot, \cdot] \right) , \tag{16}$$

*i.e., $\mathbb{R}^{n\times n}$ together with the commutator bracket $(\mathbf{A}, \mathbf{B}) \mapsto [\mathbf{A}, \mathbf{B}] = \mathbf{A}\mathbf{B} - \mathbf{B}\mathbf{A}$.*

In the context of this work, we specifically consider *quadratic matrix* Lie groups.

**Definition 2** (cf. [62]). *Given some fixed matrix $\mathbf{P} \in \mathbb{R}^{n\times n}$, the quadratic matrix Lie group $\mathcal{G}$ is defined as*

$$\mathcal{G} = \mathcal{G}(\mathbf{P}) = \left\{ \mathbf{A} \in \mathrm{GL}(n) : \mathbf{A}^\top \mathbf{P} \mathbf{A} = \mathbf{P} \right\} . \tag{17}$$

*The Lie algebra $\mathfrak{g}$ of $\mathcal{G}$ then consists of the $n \times n$ matrices $\mathbf{V}$ that satisfy $\mathbf{V}^\top \mathbf{P} + \mathbf{P}\mathbf{V} = \mathbf{0}$.*

Moreover, we consider non-autonomous linear Itô SDEs in quadratic matrix Lie groups $\mathcal{G}$. The following lemma (Lemma 3) provides sufficient conditions to ensure that an SDE solution evolves in $\mathcal{G}$. This will pave the way to introduce a class of SDEs on homogeneous spaces in Appendix E.3 via the Lie group action.

**Lemma 3** (cf. [6, Chapter 4.1], [12]). *Let $G$ be a stochastic process that solves the matrix Itô SDE*

$$\mathrm{d}G_t = \left( \mathbf{V}_0(t)\, \mathrm{d}t + \sum_{i=1}^m \mathrm{d}w_t^i \mathbf{V}_i \right) G_t , \quad G_0 = \mathbf{I}_\mathrm{n} , \tag{18}$$

*where $\mathbf{V}_0 : [0, T] \to \mathbb{R}^{n\times n}, \mathbf{V}_1, \ldots, \mathbf{V}_m \in \mathbb{R}^{n\times n}$ and $w^1, \ldots, w^m$ are independent scalar-valued Wiener processes. The process $G$ stays in the quadratic matrix Lie group $\mathcal{G}(\mathbf{P})$ defined by $\mathbf{P} \in \mathbb{R}^{n\times n}$ if*

*(i)* $\mathbf{V}_0(t) = \mathbf{K}(t) + \frac{1}{2}\sum_{i=1}^m \mathbf{V}_i^2, \quad with \quad \mathbf{K} : [0, T] \to \mathfrak{g}$, *and*

*(ii)* $\mathbf{V}_1, \ldots, \mathbf{V}_m \in \mathfrak{g}$ .

**Remark.** *Although the matrices $\mathbf{V}_1, \ldots, \mathbf{V}_m \in \mathfrak{g}$ can be chosen arbitrarily, it is natural to choose an orthogonal basis of the Lie algebra $\mathfrak{g}$. Then $m = \dim \mathfrak{g}$ and each direction in the vector space $\mathfrak{g}$ provides an independent contribution to the overall noise in Eq. (18).*

*Proof.* The process $G$ stays in the quadratic matrix Lie group $\mathcal{G}(\mathbf{P})$ if $G_t^\top \mathbf{P} G_t$ is constant, i.e., if $\mathrm{d}(G_t^\top \mathbf{P} G_t) = \mathbf{0}$. The latter is to be understood in the Itô sense, i.e.,

$$\mathbf{0} = \mathrm{d}(G_t^\top \mathbf{P} G_t) = \mathrm{d}G_t^\top \mathbf{P} G_t + G_t \mathbf{P}\, \mathrm{d}G_t + \mathrm{d}G_t^\top \mathbf{P}\, \mathrm{d}G_t$$

$$= G_t^\top \left( \left( \mathbf{V}_0^\top(t)\mathbf{P} + \mathbf{P}\mathbf{V}_0(t) \right) \mathrm{d}t + \sum_{i=1}^m \mathrm{d}w_t^i \left( \mathbf{V}_i^\top \mathbf{P} + \mathbf{P}\mathbf{V}_i \right) + \sum_{i=1}^m \mathrm{d}t\, \mathbf{V}_i^\top \mathbf{P} \mathbf{V}_i \right) G_t . \tag{19}$$

Here, $\mathrm{d}w_t^i\, \mathrm{d}w_t^j = \delta_{ij}\, \mathrm{d}t$, as $w^i$ and $w^j$ are independent scalar Wiener processes. Eq. (19) implies the following two conditions:

$$\mathbf{V}_i^\top \mathbf{P} + \mathbf{P}\mathbf{V}_i = \mathbf{0} \text{ for } i = 1, \ldots, m \quad \text{and} \quad \mathbf{V}_0^\top(t)\mathbf{P} + \mathbf{P}\mathbf{V}_0(t) = -\sum_{i=1}^m \mathbf{V}_i^\top \mathbf{P}\mathbf{V}_i . \tag{20}$$

As the following calculation shows, these conditions are satisfied if $\mathbf{V}_1, \ldots, \mathbf{V}_m \in \mathfrak{g}$ and $\mathbf{V}_0(t) = \mathbf{K}(t) + \frac{1}{2} \sum_{i=1}^{m} \mathbf{V}_i^2$ where $\mathbf{K} : [0, T] \to \mathfrak{g}$.

$$\mathbf{V}_0^\top(t)\mathbf{P} + \mathbf{P}\mathbf{V}_0(t) = \underbrace{\mathbf{K}^\top(t)\mathbf{P} + \mathbf{P}\mathbf{K}(t)}_{=\mathbf{0}} + \frac{1}{2}\left(\sum_{i=1}^{m} \mathbf{V}_i^2\right)^\top \mathbf{P} + \frac{1}{2}\mathbf{P}\left(\sum_{i=1}^{m} \mathbf{V}_i^2\right) \tag{21}$$

$$= \frac{1}{2}\sum_{i=1}^{m}\left(\mathbf{V}_i^\top\mathbf{V}_i^\top\mathbf{P} + \mathbf{P}\mathbf{V}_i\mathbf{V}_i\right) = -\frac{1}{2}\sum_{i=1}^{m}\left(\mathbf{V}_i^\top\mathbf{P}\mathbf{V}_i + \mathbf{V}_i^\top\mathbf{P}\mathbf{V}_i\right) \tag{22}$$

$$= -\sum_{i=1}^{m}\mathbf{V}_i^\top\mathbf{P}\mathbf{V}_i \ . \tag{23}$$

$\square$

**Numerical solution.** Algorithm 1 lists our choice of a numerical solver, which is the one-step geometric Euler-Maruyama scheme from [36] (see also [40, Section 3.1]).

This scheme consists of three parts. First, we divide $[0, T]$ into subintervals $[t_j, t_{j+1}]$, starting with $t_0 = 0$ until $t_{j+1} = T$. Second, we solve Eq. (18) pathwise in $\mathcal{G}$ over successive intervals $[t_j, t_{j+1}]$ by applying a *McKean-Gangolli Injection* [12] in combination with a numerical solver. Third, the (approximate) path solution is eventually computed as $G_{j+1} = G_j \exp(\Omega_{j+1})$.

---

**Algorithm 1** Geometric Euler-Maruyama Algorithm (g-EM) [36].

---

1: $t_0 \leftarrow 0$
2: $G_0 \leftarrow \mathbf{I}_\text{n}$
3: **repeat** over successive subintervals $[t_j, t_{j+1}]$, $j \geq 0$
4:     $\delta_j \leftarrow t_{j+1} - t_j$
5:     $\delta w_j^i \sim \mathcal{N}(0, \delta_j), \quad$ for $i = 1, \ldots, m$
6:     $\Omega_{j+1} \leftarrow \left(\mathbf{V}_0(t_j) - \frac{1}{2}\sum_{i=1}^{m}\mathbf{V}_i^2\right)\delta_j + \sum_{i=1}^{m}\delta w_j^i\mathbf{V}_i$          ▷ $\Omega_{j+1} \in \mathfrak{g}$.
7:     $G_{j+1} \leftarrow G_j \exp\left(\Omega_{j+1}\right)$          ▷ *Matrix exp. ensures that $G_{j+1} \in \mathcal{G}$.*
8: **until** $t_{j+1} = T$

---

**Remark.** *We remark that in the original derivation of the algorithm in [36], the group acts from the right (as in Algorithm 1). However, the same derivation can be done for a* left *group action which is what we use in our implementation and the presentation within the main manuscript. Furthermore, under condition (i) from Lemma 3, we see that* `line 6` *reduces to*

$$\Omega_{j+1} \leftarrow \mathbf{K}(t_j)\delta_j + \sum_{i=1}^{m}\delta w_j^i\mathbf{V}_i \ .$$

### E.3  SDEs in homogeneous spaces of quadratic matrix Lie groups

In the following, we use our results of Appendix E.2 to define stochastic processes in a homogeneous space $\mathbb{M}$ of a quadratic matrix Lie group $\mathcal{G}$. The main idea is, given a starting point $Z_0 \in \mathbb{M}$, to translate a path $G : [0, T] \to \mathcal{G}$ into a path $Z = G \cdot Z_0$ in $\mathbb{M}$ via the Lie group action. Notably, this construction can be done at the level of stochastic processes, i.e., a stochastic process $G : [0, T] \times \Omega \to \mathcal{G}$ induces a stochastic process $Z = G \cdot Z_0 : [0, T] \times \Omega \to \mathbb{M}$ called a *one-point motion*, see [31, Chapter 2]. In particular, if $Z_0$ follows some distribution $\mathcal{P}$ in $\mathbb{M}$ and $G$ solves an SDE of the form as in Eq. (18), then $Z$ solves $\mathrm{d}Z_t = \mathrm{d}G_t \cdot Z_0$, i.e.,

$$\mathrm{d}Z_t = \left(\mathbf{V}_0(t)\,\mathrm{d}t + \sum_{i=1}^{m}\mathrm{d}w_t^i\mathbf{V}_i\right)\cdot Z_t, \quad Z_0 \sim \mathcal{P}^{Z_0} \ . \tag{24}$$

Notably, Eq. (3) guarantees that a solution to the SDE in Eq. (18) remains in the group $\mathcal{G}$, and consequently a solution to Eq. (24) remains in $\mathbb{M}$. As we only consider Lie group actions that are given by matrix-vector multiplication, we substitute the notation $G \cdot Z$ by the more concise $GZ$ from now on.

**Remark.** *For subsequent results, it is convenient to represent Eq.* (24) *in the form of*

$$\mathrm{d}Z_t = \mathbf{V}_0(t)Z_t\,\mathrm{d}t + [\mathbf{V}_1 Z_t\,, \mathbf{V}_2 Z_t\,, \ldots, \mathbf{V}_m Z_t]\,\mathrm{d}W_t, \quad Z_0 \sim \mathcal{P}^{Z_0}\ , \tag{25}$$

*i.e., with diffusion coefficient* $\sigma : (\mathbf{z}, t) \mapsto [\mathbf{V}_1\mathbf{z}, \ldots, \mathbf{V}_m\mathbf{z}] \in \mathbb{R}^{n \times m}$ *and an* $m$-*dimensional Wiener process* $W = [w^1, w^2, \ldots, w^m]^\top$. *Notably, the matrix* $\sigma(\mathbf{z}, t)\sigma(\mathbf{z}, t)^\top$ *is given by*

$$\sigma(\mathbf{z}, t)\sigma(\mathbf{z}, t)^\top = \sum_{i=1}^{m} \mathbf{V}_i\mathbf{z}\mathbf{z}^\top\mathbf{V}_i^\top\ . \tag{26}$$

**Numerical solution.** As $Z = GZ_0$, a numerical solution to Eq. (24) is easily obtained from the numerical approximation of $G$, i.e., the numerical solution to Eq. (18) via Algorithm 1. In fact, it suffices to replace `line 2` by $Z_0 \sim \mathcal{P}^{Z_0}$ and `line 7` by $Z_{j+1} \leftarrow \exp(\Omega_{j+1})Z_j$. Notably, we do not need to compute the matrices $G_j$ but only the iterates $\Omega_j$.

### E.4 SDEs on the unit $n$-sphere

Next, we consider the homogeneous space $(\mathbb{S}^{n-1}, \mathrm{SO}(n))$, i.e., the unit $n$-sphere $\mathbb{S}^{n-1}$ together with the action of the group $\mathrm{SO}(n)$ of rotations.

Notably, the orthogonal group $\mathrm{O}(n)$ is the quadratic matrix Lie group $\mathcal{G}(\mathbf{P})$ defined by $\mathbf{P} = \mathbf{I}_\mathrm{n}$, i.e., $\mathrm{O}(n) = \left\{\mathbf{A} \in \mathbb{R}^{n \times n} \mid \mathbf{A}^\top\mathbf{A} = \mathbf{I}_\mathrm{n}\right\}$ and the special orthogonal group $\mathrm{SO}(n)$ is its connected component that contains the identity $\mathbf{I}_\mathrm{n}$. Thus, Lemma 3 ensures that a solution to Eq. (24) remains in $\mathbb{S}^{n-1}$ if $\mathbf{K} : [0, T] \to \mathfrak{so}(n)$ and $\mathbf{V}_1, \ldots, \mathbf{V}_m \in \mathfrak{so}(n)$.

In the variational setting of the main manuscript, we consider stochastic processes that solve such an SDE for the choice of $\{\mathbf{V}_i\}_{i=1}^{m} = \left\{\alpha(\mathbf{e}_k\mathbf{e}_l^\top - \mathbf{e}_l\mathbf{e}_k^\top)\right\}_{1 \le k < l \le n}$, where $\alpha \ne 0$ and $\mathbf{e}_k$ denotes the $k$-th standard basis vector of $\mathbb{R}^n$. With that choice, Eq. (24) becomes

$$\mathrm{d}Z_t = \left(\mathbf{K}(t) - \alpha^2\tfrac{n-1}{2}\mathbf{I}_\mathrm{n}\right) Z_t\,\mathrm{d}t + \alpha\,\mathrm{d}U_t Z_t, \quad Z_0 \sim \mathcal{P}^{Z_0}\ . \tag{27}$$

The diffusion term in Eq. (27) consists of a matrix-valued noise $\mathrm{d}U_t$ where $u_{ii} = 0$ and $u_{ij} = -u_{ji}$ are independent Wiener processes (for $i < j$). The following lemma (Lemma 4) converts the diffusion term to the standard form of only a vector-valued Wiener process. This facilitates (in Appendix E.5) deriving the KL-divergence between two solutions of such a SDE from existing results.

**Lemma 4.** *Let $W$ be an $n$-dimensional Wiener process and let $\mathcal{P}^{Z_0}$ be a distribution on $\mathbb{R}^n$ with finite second moment. Further, let $\mathbf{K} : [0, T] \to \mathfrak{so}(n)$ be continuous. The solutions to the SDE in Eq.* (27) *and*

$$\mathrm{d}Z_t = \left(\mathbf{K}(t) - \alpha^2\tfrac{n-1}{2}\mathbf{I}_\mathrm{n}\right) Z_t\,\mathrm{d}t + \alpha(\mathbf{I}_\mathrm{n} - Z_t Z_t^\top)\,\mathrm{d}W_t, \quad Z_0 \sim \mathcal{P}^{Z_0} \tag{28}$$

*have the same probability distribution. Furthermore, for a given initial value $Z_0 \overset{a.s.}{=} \xi \sim \mathcal{P}^{Z_0}$, a sample path solution of each equation also is a sample path solution of the other one.*

*Proof.* W.l.o.g. assume $\alpha = 1$. The SDEs from Eq. (27) and Eq. (28) have the same drift coefficient but a different diffusion term. Recall that Eq. (27) can be written as in Eq. (25), and denote the diffusion coefficients of Eq. (25) and Eq. (28) by $\sigma_1 : (\mathbf{z}, t) \mapsto [\mathbf{V}_1\mathbf{z}, \ldots, \mathbf{V}_m\mathbf{z}]$ and $\sigma_2 : (\mathbf{z}, t) \mapsto \mathbf{I}_\mathrm{n} - \mathbf{z}\mathbf{z}^\top$, respectively. The lemma then follows from [61, Theorem 2.1] which states that in the present setting, the solution to Eq. (27) and Eq. (28) have the same distribution and equal sample paths if $\sigma_1\sigma_1^\top = \sigma_2\sigma_2^\top$, i.e., if for every $\mathbf{z} \in \mathbb{R}^n$,

$$\sum_{i=1}^{m} \mathbf{V}_i\mathbf{z}\mathbf{z}^\top\mathbf{V}_i^\top = \left(\mathbf{I}_\mathrm{n} - \mathbf{z}\mathbf{z}^\top\right)\left(\mathbf{I}_\mathrm{n} - \mathbf{z}\mathbf{z}^\top\right)^\top\ . \tag{29}$$

To prove this equality, first, note that $\mathbf{I}_\mathrm{n} - \mathbf{z}\mathbf{z}^\top$ is the projection onto the orthogonal complement of $\mathbf{z}$ and so

$$\left(\mathbf{I}_\mathrm{n} - \mathbf{z}\mathbf{z}^\top\right)\left(\mathbf{I}_\mathrm{n} - \mathbf{z}\mathbf{z}^\top\right)^\top = \mathbf{I}_\mathrm{n} - \mathbf{z}\mathbf{z}^\top\ .$$

To calculate the left-hand side, we insert our choice of $\{\mathbf{V}_i\}_{i=1}^m = \big\{\alpha(\mathbf{e}_k\mathbf{e}_l^\top - \mathbf{e}_l\mathbf{e}_k^\top)\big\}_{1\leq k<l\leq n}$, which yields

$$
\sum_{i=1}^m \mathbf{V}_i\mathbf{z}\mathbf{z}^\top\mathbf{V}_i^\top = -\sum_{k=1}^n\sum_{l=1}^{k-1}\big(\mathbf{e}_k\mathbf{e}_l^\top - \mathbf{e}_l\mathbf{e}_k^\top\big)\mathbf{z}\mathbf{z}^\top\big(\mathbf{e}_k\mathbf{e}_l^\top - \mathbf{e}_l\mathbf{e}_k^\top\big)
$$

$$
= \underbrace{\sum_{k=1}^n \mathbf{e}_k\mathbf{e}_k^\top}_{\mathbf{I}_n}\underbrace{\sum_{l=1}^n \langle\mathbf{e}_l,\mathbf{z}\rangle\langle\mathbf{e}_l,\mathbf{z}\rangle}_{\|\mathbf{z}\|^2=1} - \underbrace{\sum_{k=1}^n\langle\mathbf{e}_k,\mathbf{z}\rangle\mathbf{e}_k}_{\mathbf{z}}\underbrace{\sum_{l=1}^n\langle\mathbf{e}_l,\mathbf{z}\rangle\mathbf{e}_l^\top}_{\mathbf{z}^\top}
$$

$$
= \mathbf{I}_n - \mathbf{z}\mathbf{z}^\top \ .
$$

$\square$

We want to point out that for vanishing drift $\mathbf{K} = \mathbf{0}$, the SDE in Eq. (28) becomes the well-known *Stroock* equation [23; 55] for a spherical Wiener process, i.e., a diffusion process on the sphere $\mathbb{S}^{n-1}$ that is invariant under rotations. In the variational setting of the main manuscript, we use this particular process to define an *uninformative prior*.

To illustrate that the condition $\mathbf{K} = \mathbf{0}$ defines a process $Z$ that is *invariant under rotations* $\mathbf{R} \in \mathrm{SO}(n)$, we consider the process $\mathbf{R}Z$, cf. [23, p. 220f.]. For brevity, we introduce $f : \mathbf{z} \mapsto -\alpha^2(n-1)/2\,\mathbf{z}$ and $\sigma : \mathbf{z} \mapsto \alpha(\mathbf{I}_n - \mathbf{z}\mathbf{z}^\top)$ so that Eq. (28) becomes $\mathrm{d}Z_t = f(Z_t) + \sigma(Z_t)dW_t$. Notably, the identities $\mathbf{R}f(Z_t) = f(\mathbf{R}Z_t)$ and $\mathbf{R}\sigma(Z_t)\mathbf{R}^\top = \sigma(\mathbf{R}Z_t)$ hold. Thus, the process $\mathbf{R}Z_t$ satisfies

$$
\begin{aligned}
\mathrm{d}(\mathbf{R}Z_t) &= \mathbf{R}\,\mathrm{d}Z_t \\
&= \mathbf{R}f(Z_t)\,\mathrm{d}t + \mathbf{R}\sigma(Z_t)\,\mathrm{d}W_t \\
&= \mathbf{R}f(Z_t)\,\mathrm{d}t + \mathbf{R}\sigma(Z_t)\mathbf{R}^\top\mathbf{R}\,\mathrm{d}W_t \\
&= f(\mathbf{R}Z_t)\,\mathrm{d}t + \sigma(\mathbf{R}Z_t)\,\mathrm{d}(\mathbf{R}W_t) \\
&= f(\mathbf{R}Z_t)\,\mathrm{d}t + \sigma(\mathbf{R}Z_t)\,\mathrm{d}\tilde{W}_t \ .
\end{aligned}
$$

As $\tilde{W}_t = \mathbf{R}W_t$ is again a Wiener process, the processes $Z_t$ and $\mathbf{R}Z_t$ follow the same SDE. The invariance of $Z$ w.r.t. rotations is also apparent from the SDE in Eq. (27) as $U_t$ is already invariant under rotations, i.e., at each $t \in [0,T]$ the random variables $\hat{U}_t = \mathbf{R}^\top U_t\mathbf{R}$ and $U_t$ are equal in distribution.

### E.5 Kullback-Leibler (KL) divergences

First, in Lemma 5, we state the general form of the KL divergence between path distributions of stochastic processes in $\mathbb{R}^n$.

**Lemma 5.** *Let $X$ and $Y$ be stochastic processes with (path) distributions $\mathbb{P}^X$ and $\mathbb{P}^Y$, respectively, and let $W$ be a vector-valued Wiener process; all with state space $\mathbb{R}^n$. Further, let $\mathcal{P}^{X_0}$ and $\mathcal{P}^{Y_0}$ be distributions on $\mathbb{R}^n$ with finite second moments. If $X$ and $Y$ solve[10] the following Itô SDEs on $[0,T]$,*

$$
\mathrm{d}X_t = f(X_t,t)\,\mathrm{d}t + \sigma(X_t,t)\,\mathrm{d}W_t \ , \quad X_0 \sim \mathcal{P}^{X_0} \ , \tag{30}
$$

$$
\mathrm{d}Y_t = g(Y_t,t)\,\mathrm{d}t + \sigma(Y_t,t)\,\mathrm{d}W_t \ , \quad Y_0 \sim \mathcal{P}^{Y_0} \ , \tag{31}
$$

*then the KL divergence between $\mathbb{P}^Y$ and $\mathbb{P}^X$ is given by*

$$
D_{\mathrm{KL}}\big(\mathbb{P}^Y\,\|\,\mathbb{P}^X\big) =
$$
$$
D_{\mathrm{KL}}\big(\mathcal{P}^{Y_0}\,\|\,\mathcal{P}^{X_0}\big) + \frac{1}{2}\int_0^T\int_{\mathbb{R}^n} q_{Y_t}(\mathbf{z})\,[f(\mathbf{z},t) - g(\mathbf{z},t)]^\top\,\mathbf{\Sigma}^+(\mathbf{z},t)\,[f(\mathbf{z},t) - g(\mathbf{z},t)]\,\mathrm{d}\mathbf{z}\,\mathrm{d}t \ . \tag{32}
$$

*Here, $f$ and $g$ denote vector-valued drift coefficients, and $\sigma$ denotes the matrix-valued diffusion coefficient. Furthermore, $q_{Y_t}$ denotes the marginal density of $Y_t$ (i.e., the process $Y$ at time $t$) and $\mathbf{\Sigma}^+$ denotes the pseudo inverse of $\mathbf{\Sigma} = \sigma\sigma^\top$.*

---

[10]We further assume that $f$, $g$ and $\sigma$ satisfy the assumptions of [57, Theorem 5.2.1] such that a unique solution exists (for a given $X_0, Y_0 \overset{\text{a.s.}}{=} \xi$).

For a proof via *Girsanov's theorem*, we refer the reader to [69, Section A] and [67, Lemma 2.1]. For a heuristic derivation via the limit of an Euler discretization, we refer to [43, Section 4].

The following corollary specifies Lemma 5 to a particular class of SDEs that is compatible with the SDEs introduced in Appendix E.3.

**Corollary 6.** *Let* $\mathbf{V}_0^X, \mathbf{V}_0^Y : [0,T] \to \mathbb{R}^{n \times n}$ *be continuous maps and let* $\mathbf{V}_1, \ldots, \mathbf{V}_m \in \mathbb{R}^{n \times n}$. *Further, let* $w^1, \ldots, w^m$ *be independent scalar-valued Wiener processes and let* $\mathcal{P}^{X_0}$ *and* $\mathcal{P}^{Y_0}$ *be distributions on* $\mathbb{R}^n$ *with finite second moments. If $X$ and $Y$ are stochastic processes on $[0,T]$ with (path) distributions* $\mathbb{P}^X$ *and* $\mathbb{P}^Y$, *resp., that solve*

$$\mathrm{d}X_t = \left( \mathbf{V}_0^X(t)\,\mathrm{d}t + \sum_{i=1}^m \mathrm{d}w_t^i \mathbf{V}_i \right) X_t \;, \quad X_0 \sim \mathcal{P}^{X_0} \;, \tag{33}$$

$$\mathrm{d}Y_t = \left( \mathbf{V}_0^Y(t)\,\mathrm{d}t + \sum_{i=1}^m \mathrm{d}w_t^i \mathbf{V}_i \right) Y_t \;, \quad Y_0 \sim \mathcal{P}^{Y_0} \;, \tag{34}$$

*then, the KL divergence between* $\mathbb{P}^Y$ *and* $\mathbb{P}^X$ *is given by*

$$D_{\mathrm{KL}}\left(\mathbb{P}^Y \,\|\, \mathbb{P}^X\right) = D_{\mathrm{KL}}\left(\mathcal{P}^{Y_0} \,\|\, \mathcal{P}^{X_0}\right) + \frac{1}{2} \int_0^T \int_{\mathbb{R}^n} q_{Y_t}(\mathbf{z}) \left[\mathbf{V}_0^\Delta(t)\mathbf{z}\right]^\top \mathbf{\Sigma}^+(\mathbf{z}) \left[\mathbf{V}_0^\Delta(t)\mathbf{z}\right] \mathrm{d}\mathbf{z}\,\mathrm{d}t \;, \tag{35}$$

*where* $\mathbf{V}_0^\Delta = \mathbf{V}_0^X - \mathbf{V}_0^Y$, *and* $\mathbf{\Sigma}^+(\mathbf{z})$ *denotes the pseudo inverse of* $\mathbf{\Sigma}(\mathbf{z}) = \sum_{i=1}^m \mathbf{V}_i \mathbf{z}\mathbf{z}^\top \mathbf{V}_i^\top$.

*Proof.* The noise term of the SDEs can be written as $\sigma(Z_t, t)dW_t$ where $\sigma : \mathbb{R}^n \times [0,T] \to \mathbb{R}^{n \times n}$ is specified in Eq. (25). Thus, the corollary is a direct consequence of Lemma 5, as $\sigma(\mathbf{z}, t)\sigma(\mathbf{z}, t)^\top = \sum_{i=1}^m \mathbf{V}_i \mathbf{z}\mathbf{z}^\top \mathbf{V}_i^\top$, see Eq. (26). □

Finally, the following proposition (Proposition 7) considers exactly the SDEs discussed in the main manuscript. The process $X$ is the uninformative prior process that we select in advance and the process $Y$ is the posterior process that we learn from the data. Thus, Eq. (38) is the KL-term that appears in the ELBO in our variational setting.

**Proposition 7** (KL divergence for one-point motions on $\mathbb{S}^{n-1}$). *Let* $\mathbf{K}(t) : [0,T] \to \mathfrak{so}(n)$ *be a continuous map and let* $\alpha \neq 0$. *If $X$ and $Y$ are stochastic processes on $[0,T]$ with (path) distributions* $\mathbb{P}^X$ *and* $\mathbb{P}^Y$, *resp., that solve the following Itô SDEs*

$$\mathrm{d}X_t = -\alpha^2 \tfrac{n-1}{2} X_t\,\mathrm{d}t + \alpha\,\mathrm{d}U_t X_t \qquad X_0 \sim \mathcal{P}_{\mathbb{S}^{n-1}}^{X_0} \;, \tag{36}$$

$$\mathrm{d}Y_t = \left(\mathbf{K}(t) - \alpha^2 \tfrac{n-1}{2}\mathbf{I}_n\right) X_t\,\mathrm{d}t + \alpha\,\mathrm{d}U_t Y_t \quad Y_0 \sim \mathcal{P}_{\mathbb{S}^{n-1}}^{Y_0} \;, \tag{37}$$

*where* $\mathcal{P}_{\mathbb{S}^{n-1}}^{X_0}$ *and* $\mathcal{P}_{\mathbb{S}^{n-1}}^{Y_0}$ *are distributions supported on the unit $n$-sphere $\mathbb{S}^{n-1}$ with finite second moment. Then, the KL divergence between* $\mathbb{P}^Y$ *and* $\mathbb{P}^X$ *is given by*

$$D_{\mathrm{KL}}\left(\mathbb{P}^Y \,\|\, \mathbb{P}^X\right) = D_{\mathrm{KL}}\left(\mathcal{P}_{\mathbb{S}^{n-1}}^{Y_0} \,\|\, \mathcal{P}_{\mathbb{S}^{n-1}}^{X_0}\right) + \frac{1}{2\alpha^2} \int_0^T \int_{\mathbb{S}^{n-1}} q_{Y_t}(\mathbf{z}) \,\|\mathbf{K}(t)\mathbf{z}\|^2 \,\mathrm{d}\mathbf{z}\,\mathrm{d}t \;. \tag{38}$$

*Proof.* By Lemma 4, we can replace Eq. (36) and Eq. (37) by

$$\mathrm{d}X_t = -\alpha^2 \tfrac{n-1}{2}\mathbf{I}_n\,\mathrm{d}t + \alpha\left(\mathbf{I}_n - X_t X_t^\top\right)\mathrm{d}W_t \;, \qquad X_0 \sim \mathcal{P}_{\mathbb{S}^{n-1}}^{X_0} \;, \tag{39}$$

$$\mathrm{d}Y_t = \left(\mathbf{K}(t) - \alpha^2 \tfrac{n-1}{2}\mathbf{I}_n\right)\mathrm{d}t + \alpha\left(\mathbf{I}_n - Y_t Y_t^\top\right)\mathrm{d}W_t \;, \quad Y_0 \sim \mathcal{P}_{\mathbb{S}^{n-1}}^{Y_0} \;, \tag{40}$$

as the solutions follow the same distribution $\mathbb{P}^X$ and $\mathbb{P}^Y$, respectively.

The proposition is thus a consequence of Lemma 5 for a specific choice of $f, g$ and $\sigma$, i.e.,

$$D_{\mathrm{KL}}\left(\mathbb{P}^Y \,\|\, \mathbb{P}^X\right)$$
$$= D_{\mathrm{KL}}\left(\mathcal{P}_{\mathbb{S}^{n-1}}^{Y_0} \,\|\, \mathcal{P}_{\mathbb{S}^{n-1}}^{X_0}\right) + \frac{1}{2} \int_0^T \int_{\mathbb{R}^n} q_{Y_t} \,[\mathbf{K}(t)\mathbf{z}]^\top \left(\alpha^2 (\mathbf{I}_n - \mathbf{z}\mathbf{z}^\top)(\mathbf{I}_n - \mathbf{z}\mathbf{z}^\top)^\top\right)^+ [\mathbf{K}(t)\mathbf{z}] \,\mathrm{d}\mathbf{z}\,\mathrm{d}t \;. \tag{41}$$

To finish the proof, we highlight two points. First, Lemma 4 ensures that the process $Y$ evolves on the unit $n$-sphere $\mathbb{S}^{n-1}$, so it suffices to integrate $q_{Y_t}$ over $\mathbb{S}^{n-1} \subset \mathbb{R}^n$ instead of over the entire $\mathbb{R}^n$ as in Eq. (41). Second, the integrand simplifies to

$$[\mathbf{K}(t)\mathbf{z}]^\top \left( \alpha^2 (\mathbf{I}_n - \mathbf{z}\mathbf{z}^\top)(\mathbf{I}_n - \mathbf{z}\mathbf{z}^\top)^\top \right)^+ [\mathbf{K}(t)\mathbf{z}] = \alpha^{-2} [\mathbf{K}(t)\mathbf{z}]^\top (\mathbf{I}_n - \mathbf{z}\mathbf{z}^\top) [\mathbf{K}(t)\mathbf{z}] \quad (42)$$

$$= \alpha^{-2} \|\mathbf{K}(t)\mathbf{z}\|^2 \quad , \quad (43)$$

where the first equality holds because $(\mathbf{I}_n - \mathbf{z}\mathbf{z}^\top)$ is a projection, and the second equality holds because $\mathbf{z}^\top \mathbf{K}(t)\mathbf{z} = 0$ as $\mathbf{K}(t) \in \mathfrak{g}$ is skew-symmetric. $\qquad \square$

## References (Supplementary material)

[61] E. J. Allen, L. J. S. Allen, A. Arciniega, and P. E. Greenwood. "Construction of equivalent stochastic differential equation models". In: *Stochastic Analysis and Applications* 26.2 (2008), pp. 274–297.

[4] P. Becker, H. Pandya, G. Gebhardt, C. Zhao, J. Taylor, and G. Neumann. "Recurrent Kalman Networks: Factorized Inference in High-Dimensional Deep Feature Spaces". In: *ICML*. 2019.

[62] A. M. Bloch, P. E. Crouch, J. E. Marsden, and A. K. Sanyal. "Optimal Control and Geodesics on Quadratic Matrix Lie Groups". In: *Foundations of Computational Mathematics* 8 (4 2008), pp. 469–500.

[6] R. W. Brockett. "Lie Algebras and Lie Groups in Control Theory". In: *Geometric Methods in System Theory*. Springer, 1973, pp. 43–82.

[8] N. D. Cao and W. Aziz. "The Power Spherical distribution". In: *arXiv* abs/2006.04437 (2020).

[9] F. P. Casale, A. V. Dalca, L. Saglietti, J. Listgarten, and N. Fusi. "Gaussian Process Prior Variational Autoencoders". In: *NeurIPS*. 2018.

[10] R. T. Chen, Y. Rubanova, J. Bettencourt, and D. Duvenaud. "Neural Ordinary Differential Equations". In: *NeurIPS*. 2018.

[12] G. S. Chirikjian. "Stochastic models, information theory, and Lie groups". In: Applied and numerical harmonic analysis. Boston, Basel, and Berlin: Birkhäuser, 2012, pp. 361–388.

[18] H. Fu, C. Li, X. Liu, J. Gao, A. Celikyilmaz, and L. Carin. "Cyclical Annealing Schedule: A Simple Approach to Mitigating KL Vanishing". In: *NAACL*. 2019.

[23] K. Itô. "Stochastic Calculus". In: *Lect. Notes Phys.* 39 (1975). Ed. by H. Araki, pp. 218–223.

[63] I. Karatzas and S. E. Shreve. *Brownian motion and stochastic calculus*. 2nd edition. Springer, 1998.

[28] D. P. Kingma and J. Ba. "Adam: A Method for Stochastic Optimization". In: *ICLR*. 2015.

[64] P. Kloeden and E. Platen. *Numerical methods for stochastic differential equations*. 1991.

[65] J. M. Lee. *Introduction to smooth manifolds*. Graduate texts in mathematics. Springer, 2003.

[30] X. Li, T.-K. L. Wong, R. T. Chen, and D. Duvenaud. "Scalable Gradients for Stochastic Differential Equations". In: *AISTATS*. 2020.

[31] M. Liao. *Levy processes in Lie groups*. Providence, R.I.: Cambridge University Press, 2004.

[36] G. Marjanovic and V. Solo. "Numerical Methods for Stochastic Differential Equations in Matrix Lie Groups Made Simple". In: *IEEE Transactions on Automatic Control* 63.12 (2018), pp. 4035–4050.

[40] M. Muniz, M. Ehrhardt, M. Günther, and R. Winkler. "Higher strong order methods for linear Itô SDEs on matrix Lie groups". In: *BIT Numerical Mathematics* 62.4 (2022), pp. 1095–1119.

[43] M. Opper. "Variational Inference for Stochastic Differential Equations". In: *Annalen der Physik* 531.3 (2019), p. 1800233.

[66] D. Revuz and M. Yor. *Continuous Martingales and Brownian Motion*. 3rd edition. Springer, 1999.

[48] Y. Rubanova, R. T. Chen, and D. Duvenaud. "Latent ODE for Irregularly-Sampled Time Series". In: *NeurIPS*. 2019.

[67] D. Sanz-Alonso and A. M. Stuart. "Gaussian approximations of small noise diffusions in Kullback-Leibler divergence". In: *Communications in Mathematical Sciences* 15.7 (2017), pp. 2087–2097.

[68]  S. Särkkä and A. Solin. *Applied stochastic differential equations*. Cambridge University Press, 2019.

[49]  M. Schirmer, M. Eltayeb, S. Lessmann, and M. Rudolph. "Modeling Irregular Time Series with Continuous Recurrent Units". In: *ICML*. 2022.

[50]  S. N. Shukla and B. M. Marlin. "Multi-Time Attention Networks for Irregularly Sampled Time Series". In: *ICLR*. 2021.

[51]  I. Silva, G. Moody, D. J. Scott, L. A. Celi, and R. G. Mark. "Predicting in-hospital mortality of ICU patients: The PhysioNet/Computing in cardiology challenge 2012". In: *Computing in Cardiology* 39 (2012), pp. 245–248.

[69]  T. Sottinen and S. Särkkä. "Application of Girsanov theorem to particle filtering of discretely observed continuous-time non-linear systems". In: *Bayesian Analysis* 3.3 (2008), pp. 555 –584.

[55]  D. W. Stroock. "On the growth of stochastic integrals". In: *Zeitschrift für Wahrscheinlichkeitstheorie und Verwandte Gebiete* 18.4 (1971), pp. 340–344.

[57]  B. Øksendal. *Stochastic Differential Equations*. Universitext. Springer, 1995.

[60]  Ç. Yildiz, M. Heinonen, and H. Lahdesmäki. "ODE$^2$VAE: Deep generative second order ODEs with Bayesian neural networks". In: *NeurIPS*. 2019.

