# OpenReview forum: "Latent SDEs on Homogeneous Spaces"
_NeurIPS.cc/2023/Conference — NeurIPS 2023 poster_

### Official Review · Reviewer_uKNv · 2023-06-14

**Soundness:** 3 good
**Presentation:** 1 poor
**Contribution:** 3 good
**Rating:** 6
**Confidence:** 3

**Summary:**

In this paper the authors develop machinery for performing variational inference on latent functions in models where observations are generated by a latent stochastic process.  That is, the generative model is that a path in some latent space is generated according to a prior, and then we observe (noisy) values of this path at a subset of the points.  In principle this paper deals with a general case where the latent space is a "homogeneous space" which is a manifold such that there is a Lie group such that for any pair of points in the manifold one can translate one of the points to the other using a single element of the Lie group.  In practice, the paper focuses on the specific case of the manifold being an n-sphere (so that the Lie group is the rotations, SO(n)), and the generative model is that the initial point is sampled uniformly from the sphere and then evolves according to a (scaled) Brownian motion on the sphere.  Once the authors develop their inference machinery, they apply their method to a number of regression, classification, interpolation, and extrapolation problems, showing that this latent SDE formulation can be useful in some settings.

**Strengths:**

The proposed method is very elegant and the mathematics is nice.  Moving to SDEs on a compact latent space allows for many nice features like being able to define uninformative priors and having nice correspondences between the initial distribution and the stationary distribution of the SDE.

The simplicity and performance on the considered tasks is strong motivation for the usefulness of the construction, and having access to posteriors over entire latent trajectories allows for a number of interesting tasks like interpolation.

**Weaknesses:**

- The main weakness for me is the presentation of the material.  I will try to provide more specific feedback in the following points, but overall there was a large disconnect between the setup in equations (1) and (2) and then the actual applications.  Specifically, (2) is listed as the objective function, but then it's not clear what the objective function would be for the regression and classification tasks.  The classification task presumably has a different generative model than the one presented between equations (1) and (2), and it's not clear where the class label would go, and what data is used where.  What parts of the models are learnable?
-  The introduction is a bit confusing.  For example, it is unclear while reading the introduction what is path-valued and what is not.  Line 105 says everything is path-valued but then lines 93-94 say that we only have a finite number of discrete observations of the path.
- I found the condition on $h$ in line 103 confusing.  Is this at all a restriction?  It seems like equation (1) automatically makes $X_t | Z_t$ normally distributed with mean $\mu_t = h(Z_t)$ and variance $\mathbf{R}$.
- I defer to the authors on how they want to present it, but I was confused by the presentation of the ELBO around (2) because in general the KL divergence between distributions over functions determined by SDEs will be infinite unless the diffusion terms are the same.  The authors address this thoroughly and clearly later in the manuscript, but sweep this (to me) important point under the rug in the introduction.
- Equation (12) and Figure 2 make it clear that the inference network takes in the $\mathbf{x}$'s and then learns the hyperparameters of the initial distribution and the SDE.  Is the inference network necessary because of the sample sizes considered here?  Wouldn't it be possible to just directly optimize the set of $\mathbf{K}_i^\phi$'s that specify the SDE (and the parameters of the initialization) for each set of observations?  Some comment on the distinction of what is being used for amortization (and why) vs. modeling flexibility would be useful.
- The notation around equation (7) is very confusing to me.  It appears that the time points $t_k$ represent the time that has passed since time $0$, but then it feels like $Z_{t_j}$ should be $G(t_j)G^{-1}(t_{j-1})Z_{t_{j-1}}$.  Perhaps relatedly, it's not obvious to me what the connection is between $G(t_j)$ and the SDE in (4). In particular it feels like to solve the SDE for the interval $[t_{j-1}, t_j]$ you would need a different initial condition (the distribution over $G(t_{j-1})$ obtained by solving the SDE up to that point.  Apologies if I'm being slowing here.
- The example of extrapolating the rotated MNIST was a bit confusing.  Doesn't the extrapolation rely exclusively on the Chebyshev polynomials behaving well after the end of the interval over which they are trained?  Relatedly, how is the end time point chosen during the VI optimization and does it matter?  The KL term seems sensitive to how long of an interval is considered.  E.g., if we consider data sampled on [0, 1], but then compute the KL on paths from [0, T] with T>>1, we would want the drift terms to eventually relax back to matching the prior.
- Lines 371-372 are confusing: what is meant by saying that only the initial time point is observed?  The prior is driftless, so it feels like if one just observes the initial state it should be hard to get directional rotations using the latent SDE model.

Typos:
- "on a various time series interpolation" --> "on various time series interpolation"
- "the paradigm of Parameterizing the vector fields" --> "the paradigm of parameterizing the vector fields"
- Line 171: "in context of" --> "in the context of"
- Line 186: "are show in" --> "are shown in"
- Line 189: "This allows to select" --> "This allows selecting"
- I believe the equation prior to line 192 is missing a square on the Frobenius norm term since the vector norm on the lefthand side is squared
- Line 265: "with label switches need" --> "with label switches needing"
- Line 358: "Upon receiving first" --> "Upon receiving the first"
- I believe equation (35) has an erroneous $\mathbf{z}$ on the lefthand side (the righthand side is matrix valued)

**Questions:**

The actual generative model is quite simple, and very little of it is learned.  In particular, the prior distribution has only a single learnable parameter.  Presumably most of the modeling flexibility comes from learning the mapping from the latent space to the observable data (i.e., $p(\mathbf{x}(t) | \mathbf{z}(t))$).  Is there a reason for not learning a drift term for the prior?  Would one run into identifiability issues with more complex priors?

How difficult would it be to extend the implementation and framework to other homogeneous spaces?  Are there any applications where such a generalization would be obviously useful?

**Limitations:**

The authors have adequately addressed potential social implications.

---

> ### Author Rebuttal · Authors · 2023-08-08
>
> Below, we address all points and outline how we will revise our manuscript. If there are further questions, we are more than happy to answer.
> > ... (2) is listed as the objective function, but then it's not clear what the objective function would be for the regression and classification tasks ... What parts of the models are learnable?
>
> We kindly refer to our **General Response** (♦) to clarify the optimization objective & learnable model components.
> > ... it is unclear while reading the introduction what is path-valued and what is not. Line 105 says everything is path-valued but then lines 93-94 say that we only have a finite number of discrete observations of the path.
>
> In the mentioned paragraph, we wanted to express that our setting differs from a typical VAE setting in that the random variables are stochastic processes, i.e., path-valued. To clarify, in `l105`, "observations" refer to realizations $\mathbf x = X(\omega):[0,T]\to \mathbb R^d$ of the process $X$. In practice, not the entire $\mathbf x$ is available, but only a finite tuple $(\mathbf x(t_1), \dots, \mathbf x(t_m))$ of evaluations of $\mathbf x$. We will clarify this in a final version.
> > I found the condition on $h$ in line 103 confusing. Is this at all a restriction? It seems like equation (1) automatically makes $X_t|Z_t$
>  normally distributed with mean $\mu_t = h(Z_t)$ and variance $\boldsymbol{R}$.
>
> Overall, Eq. (1) in our "Preliminaries" section is unnecessary and (apparently) caused confusion; please see our **General Response** (♥).
>  > I was confused by the presentation of the ELBO around (2) because ...
>
> We agree that even at this (early) point in the manuscript, a remark would help to clarify that when distributions are determined by SDEs, the diffusion terms need to agree to avoid infinite KL divergence. We will add a remark.
> > Equation (12) and Figure 2 make it clear that the inference network takes in the $\mathbf x$'s and then learns the hyperparameters of the initial distribution and the SDE. Is the inference network necessary because of the sample sizes considered here? ...
>
> The inference network learns a mapping from observations to the initial state parameterization of the SDE and to the coefficients in $\mathbf K^\phi_i$. In principle, these parameters could be optimized directly and separately for each time series, but this would not scale well: in fact, at test time, each new sequence of observations would require optimizing over the SDE parameterization again. With an *inference network*, one simply needs one forward pass through the model.
> > The notation around equation (7) is very confusing to me ...
>
> The reviewer is right that the notation of $G(t_j)$ around Eq. (7) is inconsistent with the SDE on $G_t$ in Eq. (4) and that $G(t_j)$ needs to be replaced with $G_{t_j} G_{t_{j-1}}^{-1}$, i.e., with $\exp(\Omega_{j})$ from Alg. 1 in the suppl. material. In this respect, for $t \in [t_{j-1}, t_j]$, the SDE in Eq. (4) is to be understood as an SDE on $G_t$ with initial value $G_{t_{j-1}}$, or, equivalently, as an SDE on $G_t G_{t_{j-1}}^{-1}$ starting at the identity. We will clarify and update the notation.
> > The example of extrapolating the rotated MNIST was a bit confusing. Doesn't the extrapolation rely exclusively on the Chebyshev polynomials ...
>
> For extrapolation, we use the *same* model that was trained on the interpolation task, which has only seen data in the time $[0,1]$ (i.e., the 1st full rotation, w/o knowledge of extrapolation times $t>1$). Accordingly, the path KL div. is only computed on $[0,1]$. At test time, predictions at future time points are generated by integrating over a longer time range. It is true that extrapolation quality primarily depends on how well the drift term behaves for $t>1$. To verify the intuition that a constant velocity on the sphere is a suitable model for the constant rotation velocity in the data, we use only the first $K=1$ Chebyshev polynomial, i.e., *a constant*. $K>1$ yields better results but at a loss of extrapolation quality as the higher-order polynomials are less well-behaved for large $t$.
> > Lines 371-372 are confusing: what is meant by saying that only the initial time point is observed? ...
>
> We agree that the phrase "only the initial time point is observed" is misleading: for *loss computation*, we **do** have information about 11 out of 16 images rotated by multiples of 22.5°. However, for training & testing, only the initial upright '3' is available as model **input**. This strategy works as the rotation speed in the data is constant. For rotation speeds that vary across time series, we would likely need more input images.
> > ... Is there a reason for not learning a drift term for the prior? Would one run into identifiability issues with more complex priors? ...
>
> We use an uninformative prior with zero drift to express our limited knowledge about the underlying latent dynamics. In case of additional information, learning a drift component for the prior might be beneficial, e.g., learning a prior with constant velocity might be a good idea on Rotated MNIST.
> Regarding identifiability, note that we are not interested in interpreting the parameters of the latent SDE (in isolation) but only in its utility for downstream tasks.
> Identifiability would be more relevant when a specific parametric SDE is prescribed (e.g., from a physical model) and one is interested in the values of the fitted parameters.
> > How difficult would it be to extend the implementation and framework to other homogeneous spaces? Are there any applications where such a generalization would be obviously useful?
>
> For more modeling flexibility, SDEs on $\mathbb R^n$, induced by matrix multiplication with $G\in GL(n)$, can be used. We already implemented this variant, and **R-Tbl. 1** in the attached PDF lists results on Rotated MNIST.  Another application (cf. reviewer XRoq) is latent dynamics in a hyperbolic space induced by $O(1,n)$, e.g., for modeling relativistic dynamics.

---

> > ### Comment · Reviewer_uKNv · 2023-08-10
> >
> > Thank you very much for the thorough response.  You have substantially clarified things, and I will increase my score correspondingly, under the assumption that these clarifications will make it into the next version of the paper.
> >
> > I also think it would be good to include something akin to your response here regarding the extrapolation on MNIST.  It feels like this particular example is leveraging information that will not be true in general applications, and so presents a somewhat overly rosy view of how this approach is expected to extrapolate on arbitrary real world applications.
> >
> > Thank you again.

---

### Official Review · Reviewer_XRoq · 2023-07-05

**Soundness:** 3 good
**Presentation:** 4 excellent
**Contribution:** 3 good
**Rating:** 7
**Confidence:** 4

**Summary:**

This paper deals with the problem of variational inference for sequential data, i.e. time series, using latent SDEs. The idea of this class of methods is to assume the stochastic process that is observed is related to a generative SDE in a latent space, whose parameters need to be inferred from data in a Bayesian fashion. The novelty of this paper is to consider SDE priors that live on homogeneous spaces, and in particular on the sphere (acted upon transitively by matrix multiplication by $SO(n)$). Many works have taken the path of having either a spherical or sequential latent space recently, but have never considered latent SDEs on the sphere as priors for variational inference of the posterior process. This is made possible by designing an uniformative sequential prior process on the sphere (with a tractable KL divergence) and a parametric class of posterior SDEs that provably lead to solutions on the sphere, and using an associated geometric Euler Maruyama scheme for the numerical integration of the SDE. The model is then optimized using an ELBO loss that is adapted to sequential variables and the SDE at hand, using the Power Spherical law as a reparametrization-friendly prior distribution on initial condition, and furthermore fitting the drift of the latent SDE. The method is then tested on a number of datasets and machine learning tasks, for which having both a sequential and spherical prior may or may not be a natural choice. Results indicate that the proposed method is at least comparable to other SOTA approaches on those problems on the tasks adressed.

**Strengths:**

Strengths :
- The theory of ODEs on Lie Groups and the associated geometric integration schemes is nicely put to use, and is potentially applicable to any homogeneous space. It would be quite nice to try and extend all this work to hyperbolic spaces, which have been used a lot in ML recently, and happen to also be homogeneous spaces.
- I found the paper well written and enjoyable to read.
- The example of the sphere is compelling and leads to the specification of stochastic processes defined through SDEs on the sphere both for the prior and posterior.  The tools introduced lead to a new prior distribution for stochastic processes on the sphere, and tractable posterior distributions shown to be usable in a variational inference context.
- An extensive experimental study is carried out on different standard datasets. The proposed framework is applicable to a number of different tasks involving time series : classification (of each time step or of the global sequence), interpolation, regression. The method proves competitive with respect to SOTA approaches for each task.



**Weaknesses:**

Weaknesses :

- As often in works involving spherical, or more generally hyperbolic latent space, the motivation of using them is not always crystal clear for all applications. For data involving some kind of periodicity, as most of those tested here (pendulum, rotating MNIST), this makes sense, but for others it is not so obvious why one would want to use spherical latent spaces. Would this method work for e.g. time series prediction of chaotic data, e.g. the Lorenz system ? However this is a minor remark. Once the need for a spherical latent space has been established, the proposed work extends successfully variational inference for sequential data to this setting.

- I find it a bit disappointing not to find an application of the proposed method to uncertainty quantification, or at least one where the capacity to sample from the posterior process is not put to use. To me, this is the main interest of working with latent SDEs instead of ODEs : one gets a sequential generative model capable of sampling, computing expectations, covariance or other statistics. This could be showcased e.g. through a simple experiment of forecasting time series, showing predictions and confidence intervals.


**Questions:**

- Are the error bars reported in the tables computed though multiple training sessions or simply several samples of the posterior, for a given trained model ?

- On a related note, I am not sure how the model is adapted for tasks such as per-frame classification. I get that the purpose of the h function is to potentially map to another space, e.g. using a softmax for classification problems, and I imagine the second term of the ELBO is the one that needs to be adapted depending on the model. It could be useful to detail one example (e.g. classification) and explaining what eq. (2) becomes in that case. Could h be learned jointly with the posterior instead of being assumed to be known ?

- In figure 1, I would have been interested in seeing a few trajectories of posterior samples that do not have a constant label, to see if the proposed intuition for the trajectories still holds in that case.

- How does the neural net used to fit the parameters of the initial condition given by the spherical power law constrain them to be positive for the concentration parameter and on the sphere for the location parameter ?

- Note that Figure 2 does not display properly on all the different pdf readers I tried. For one of them, I cannot see the boxes and arrows, but just the text, which makes the figure hard to read.

**Limitations:**

- One of the main limitations of the work is the lack of motivation of using SDEs/generative models instead of simply ODEs/deterministic models, if the sampling capacity of the model is never really put to use, while in many domains involving time series uncertainty quantification is a key scientific issue (see above for detailed comments on this point).

- Another minor limitation is the limitation of a geometric Euler Maruyama solver for the SDE, while more accurate solvers (of Runge Kutta Munthe Kaas type) exist. Those are not used so as not to end up with a too much computational burden, but I wonder how much of a problem this would be. More generally, I also would like to have an idea of the required computational time/complexity of the proposed model, and how much it would depend on the used solver.

---

> ### Author Rebuttal · Authors · 2023-08-08
>
> First, we thank the reviewer for the positive feedback! The suggestion to extend our method to hyperbolic spaces $\mathbb H^n$ is quite interesting and opens up a new direction of future applications that we did not think of so far, e.g., relativistic dynamics in Minkowski space-time. The hyperboloid model of $\mathbb H^n = O(1,n)/(O(1) \times O(n))$ enables defining a stochastic process on $\mathbb H^n$ via an SDE in $O(1,n)$ or its identity component $SO^+(1,n)$, similar to the presented case of $\mathbb S^n$ and $O(n)$. In fact, $O(1,n)$ is also a quadratic Lie group and nicely fits into our setting.
>
> > ... Would this method work for e.g. time series prediction of chaotic data, e.g. the Lorenz system? ...
>
> We fully agree that the choice of latent space geometry should be informed by the data. For data that involves some kind of periodicity, a spherical latent space is a good choice, but for chaotic data, this is less clear. To investigate the reviewer's suggestion regarding the Lorenz system, we used our latent dynamic model to replicate the experimental setup of [Xi et al., 2020] for fitting a *stochastic* Lorenz attractor. As can be seen from **R-Fig. 1** in the attached PDF (showing 75 posterior samples), our method is expressive enough to model the dynamics of this system. As a side remark, a second motivation for using a spherical or, more generally, a homogeneous latent space is that it enables using Lie group solvers with stronger convergence guarantees (see [Muniz et al., 2022], [Marjanovic et al., 2018]).
>
> > I find it a bit disappointing not to find an application of the proposed method to uncertainty quantification, or at least one where the capacity to sample from the posterior process is not put to use ...
>
> We agree with the reviewer that leveraging the capability of the SDE model for uncertainty quantification would undoubtedly be very interesting. In the current work, we primarily focused on developing the methodological foundation and limited our experiments to a thorough evaluation of interpolation and regression/classification capabilities on existing benchmark data. A similarly extensive evaluation of uncertainty quantification seemed out of scope. Nevertheless, following the reviewer's suggestion, **R-Fig. 2** in the attached PDF now includes a visualization of uncertainty in the angle predictions of the **Pendulum angle regression** experiment (for one testing instance). While this is a regression and not a forecasting problem, it highlights XRoq's point of the model's capability to assess uncertainty. Qualitatively, uncertainty is higher in regions where the angle prediction is less accurate. We will include this figure (and additional visualizations of this kind) in a final version and will more prominently point out the possibility for uncertainty assessment.
>
> > Are the error bars reported in the tables computed through multiple training sessions or simply several samples of the posterior, for a given trained model?
>
> The error bars in the tables (e.g., 0.5e-3 for pendulum regression) are standard deviations computed over **5** training runs (with different random seeds). The standard deviations with respect to sampling from the posterior are lower (0.05e-3). We will make this clear in a final version.
>
> > On a related note, I am not sure how the model is adapted for tasks such as per-frame classification ...
> > Could h be learned jointly with the posterior instead of being assumed to be known?
>
> We apologize for the confusion and refer the reviewer to our **General Response** section for full clarification. In short, $h$ can be removed from the preliminaries section, as our variational setting can be described without it. We intended to introduce the observation model in liaison with the more common approach of simply stating the generative model, but this caused more confusion than it helped.
>
> Regarding a detailed description of our setting on one example: let's take the pendulum angle regression task: in that case, the decoder part of our model (1) yields $p(\mathbf{x}(t)|\mathbf{z}(t))$ modeled as a Gaussian distribution with its mean representing the reconstructed observation (i.e., pendulum images) and (2) an additional neural network (as in [Schirmer et al., 2022]) maps latent states $\mathbf{z}(t)$ to the pendulum's angle. Deviations from the desired ground truth angle are measured via an MSE loss. Hence, the overall optimization objective becomes *MSE + negative ELBO*.
>
> >In figure 1, I would have been interested in seeing a few trajectories of posterior samples that do not have a constant label, to see if the proposed intuition for the trajectories still holds in that case.
>
> Following your suggestion, we selected two trajectories with constant class labels and **one** trajectory with a label switch. **R-Fig. 4** (right) in the attached PDF shows that the latter has a larger drift component (as it needs to cross decision boundaries). The coloring indicates the *predicted* class label (with a 30\% error in the label-switch case and 0\% error in the constant-label case). Quantitatively, **R-Fig. 4** (left) shows the distribution of path KL divergences *with* and *without* label switches, highlighting that trajectories *with* label switches indeed deviate more from the driftless prior.
>
> > How does the neural net used to fit the parameters of the initial condition given by the spherical power law constrain them to be positive for the concentration parameter and on the sphere for the location parameter?
>
> To ensure positivity of the concentration parameter ($\kappa$) of the power-spherical distribution, we take the square. Initially, we experimented with an exponential mapping but found that to be too aggressive. The location parameter is divided by its norm.
>
> > Note that Figure 2 does not display properly on all the different pdf readers I tried ...
>
> Thank you for pointing this out. We will obviously fix this issue in a final version.

---

### Official Review · Reviewer_tqmQ · 2023-07-06

**Soundness:** 4 excellent
**Presentation:** 4 excellent
**Contribution:** 3 good
**Rating:** 7
**Confidence:** 2

**Summary:**

The authors are interested in learning neural SDE models. Instead of parameterising arbitrary latent SDEs, the authors restrict their attention to homogeneous spaces, and in particular the unit sphere, in order that they can leverage the transitive group (the Lie group) to construct an SDE in the space in terms of an SDE in the Lie group, whose logarithm is a linear SDE, which leads to convenient solutions. The major advantage of this is using a straightforward discretise-then-optimise approach without an explosion of computational cost. They perform a thorough evaluation against other neural ODE/SDE methods, and show competitive performance despite the more restrictive form of the latent SDE.

Post-discussion: The major concern has been addressed. The authors have pledged to extend their evaluation of the efficiency of their method to other experiments in the paper, which will be sufficient to address that part.

**Strengths:**

1. (major) To my knowledge the approach is novel and markedly different to other latent SDE methods, but is applicable to many of the same tasks, so is highly relevant.
2. (major) The evaluation is generally very thorough (though see weaknesses), with a wide range of related methods and different tasks evaluated.
3. (major) The code is available, which improves reproducibility, and is based on a widely used framework (pytorch) which improves possible impact.
4. (minor) The paper is quite clearly written, including good discussion of the empirical findings.

**Weaknesses:**

1. (major) The method is claimed to be efficient for learning, but no evidence is provided to support this claim -- the experiments as they are demonstrate that the method can produce a test performance which is competitive with apparently more flexible approaches, and that the relative performances may depend somewhat on the task. For the reader to judge how far this method is more efficient, a quantitative evaluation of the time for learning for different models would help.

**Questions:**

typos:


* lines 25, 58: Paramaterizing -> parameterizing
* line 178: extra $≤$
* line 180: arithmetics -> arithmetic

**Limitations:**

Limmitations are well discussed explicitly in the paper.

---

> ### Author Rebuttal · Authors · 2023-08-08
>
> >(major) The method is claimed to be efficient for learning, but no evidence is provided to support this claim -- the experiments as they are demonstrate that the method can produce a test performance which is competitive with apparently more flexible approaches, and that the relative performances may depend somewhat on the task. For the reader to judge how far this method is more efficient, a quantitative evaluation of the time for learning for different models would help.
>
> We agree that additional empirical evidence is required to demonstrate and quantify that our method is more efficient. We will add a subsection to discuss computational/runtime aspects.
>
> Importantly, during the author's response period, we already ran a careful runtime comparison on the Rotated MNIST interpolation task, with results providing evidence in support of our claim. We kindly refer the reviewer to our **General Response** section for a detailed discussion and, in particular, to **R-Tbl. 1** in the attached PDF. Overall, our method is on par in terms of runtime with a latent ODE model of comparable size (#parameters) and substantially faster than the conceptually closer (and more flexible) latent SDE approach of [Xi et al., 2020], both evaluated using an Euler(-Maruyama) scheme.
>
> We also like to thank the reviewer for pointing out typos in the manuscript.

---

> > ### Comment · Reviewer_tqmQ · 2023-08-10
> >
> > Thanks, this is helpful to see. Just a few short follow ups.
> >
> > 1. Will you produce similar results for the other datasets?
> > 2. For R Fig 3, could you include the uncertainty estimates? (since you have run with five initialisations)
> > 3. It seems like the main efficiency you gain is vs other latent SDE methods, but maybe the motivation for using SDE models over ODE models is not so clear in the current manuscript, but I think this is already satisfactorily addressed in your response to reviewer XRoq.

---

> > > ### Author Response · Authors · 2023-08-10
> > >
> > > First, we thank the reviewer for the prompt response.
> > >
> > > **ad 1)** Yes, we will produce similar results for the other datasets in a final version (assuming the reviewer is referring to the runtime experiments).
> > >
> > > **ad 2)** We replotted **R-Fig. 3** over all 5 runs, with very little runtime variation per approach. Unfortunately, we cannot update the attached PDF at this point, but we will include such a figure in a final version. Specifically, we will show the mean across all runs and shade the standard deviation (as error bars are hard to see due to the large number of points on the $x$-axis.)
> > >
> > > **ad 3)** It is correct that our main efficiency gain is wrt. other SDE methods. Following your (and Xroq's) suggestion, we will update our manuscript to more prominently point out the utility of being able to sample from the (latent) posterior process, e.g., in the context of uncertainty assessment.

---

> > > > ### Comment · Reviewer_tqmQ · 2023-08-14
> > > >
> > > > Thanks, this sounds promising, and addresses the main issues I have raised. Although we will not be able to evaluate the significance of the efficiency improvement during the discussion phase, the results presented so far are sufficiently reassuring. I will update the review and score in the coming days.

---

> > > > > ### Author Response · Authors · 2023-08-20
> > > > >
> > > > > Dear reviewer tqmQ,
> > > > >
> > > > > thank you for your positive feedback! This is a polite reminder that the discussion period between authors and reviewers will end very soon. If you are still convinced to increase your score, we would appreciate your effort to update your review.
> > > > >
> > > > > Sincerely
> > > > > The authors

---

> > > > > > ### Comment · Reviewer_tqmQ · 2023-08-20
> > > > > >
> > > > > > It is done

---

### Official Review · Reviewer_GYzQ · 2023-07-23

**Soundness:** 3 good
**Presentation:** 3 good
**Contribution:** 3 good
**Rating:** 7
**Confidence:** 3

**Summary:**

The paper provides an affirmative answer to a very natural and intriguing question: Can we simplify the underlying latent model describing the dynamics of a temporal process so that it can overcome the computational and technical challenges with neural ODEs/SDEs while accurately modeling the real-world phenomenon? The authors propose SDE models that arise from the action of a Lie group on a homogeneous space instead of an arbitrary SDE. They also demonstrate at-par performance of their approach on benchmark regression, classification and interpolation problems when compared to the existing methods.


**Strengths:**

This work is really interesting as it opens a new direction of research that could reduce the model's degrees of freedom and yet achieve (or nearly) the sota. It may encourage machine learners to leverage recent developments in the SDE literature and further simplify learning a time series phenomenon. The paper is well written.

**Weaknesses:**

1. The paper claims in the introduction that their approach significantly reduced computing efforts.
                    “ However, .......... computing gradients.”
However, there is no comparison with other approaches I could find in the numerical section. Also, a mathematical discussion on the reason behind computational gain is absent.

2. I think the paper would benefit from adding more discussion on the statement below in the main paper.

“Numerical solutions to such an SDE are computed with a simple one-step geometric Euler-Maruyama scheme for which the “discretize-then-optimize” strategy of backpropagating gradients during learning is not a limiting factor."


**Questions:**


1. Why do we need such a specific SDE form for G_t in the display (4)?
2. Why do we not need a reparametrization trick for the proposed approach? I found no discussion on this in the paper (except a standard comment in line 111).

Minor.
You followed [47] (in the paper)or [48](in the appendix) for preprocessing the human activity dataset. Which one did you follow?

**Limitations:**

The paper includes a discussion on the limitations and possible societal impact.

---

> ### Author Rebuttal · Authors · 2023-08-08
>
> Reviewer GYzQ identified two weaknesses in our submission that we will address below.
>
> > The paper claims in the introduction that their approach significantly reduced computing efforts ...
>
> To address the remark on significantly reduced computing effort, we reran our experiments on the rotated MNIST dataset and carefully compared the training time of our method to the latent ODE and SDE models from [Rubanova et al., 2019] and [Xi et al., 2020], respectively. We report the results in **R-Tbl. 1** & **R-Fig. 3** of the attached PDF, and refer the reviewer to our **General Response** section for a detailed discussion.
>
> >I think the paper would benefit from adding more discussion on the statement below in the main paper.
> >“Numerical solutions to such an SDE are computed with a simple one-step geometric Euler-Maruyama scheme for which the “discretize-then-optimize” strategy of backpropagating gradients during learning is not a limiting factor."
>
> The word choice in this paragraph is somewhat unfortunate. The phrasing is too general and should rather refer to the actual use case scenario. We will adjust this accordingly in the final version. As we need to backpropagate gradients through each solver step, memory complexity scales linearly with the number of (fixed) time steps multiplied by the amount of memory consumed per step (which depends on the latent space dimensionality). Our statement in the manuscript should be understood in the context of the **latent** dynamics setting, where typically, a low dimensionality of the latent space is expected to suffice for good performance on downstream tasks (as shown in our experiments). While each step of the solver is quite simple (due to the construction of our SDE), there may well be limitations in using the “discrete-then-optimize” strategy once the latent space dimensionality reaches a certain point. We will rephrase (i.e., tone down) this statement and add more discussion on that issue.
>
> >Why do we need such a specific SDE form for $G_t$ in the display (4)?
>
> We want to clarify that we do not state that this form of an SDE for $G_t$ is necessary. We restrict our SDEs to the form as in Eq. (4) because it constitutes an easy and natural way to define a stochastic process $G_t$ that evolves in the Lie group in terms of drift and diffusion residing in the Lie algebra. Moreover, for SDEs of this form, we can rely on numerical solvers from [Marjanovic et al., 2018]. On the tasks and datasets considered in our work, SDEs of such a form appear to be sufficient to achieve competitive performance (as noted by the reviewer). Nevertheless, the formulation in Eq. (4) is quite general. The restrictions on the coefficients $V_0,V_1,\dots,V_m$ specified in Eq. (5) only state that $V_1,\dots,V_m$ are in the Lie algebra and that $V_0$ is determined by an additional element in the Lie algebra that needs to be adjusted by a correction term to cancel out the stochastic drift away from the Lie group.
>
> > Why do we not need a reparametrization trick for the proposed approach? I found no discussion on this in the paper (except a standard comment in line 111).
>
> We apologize for not being clear enough on this point. The short answer is 'yes'; we do need a reparametrization trick. To be more specific, sampling from our posterior (path-) distribution on the sphere consists of two steps. First, we sample an initial value from a power-spherical distribution with location parameter $\boldsymbol{\mu}$ and concentration parameter $\kappa$. Importantly, this distribution allows for a reparametrization trick which we use for training. Second, we numerically solve an SDE in the homogeneous space that starts at the sampled initial value with a geometric Euler-Maruyama scheme. This also uses a reparametrization trick, as with each update step, a random matrix with elements $\sim \mathcal N(0,1)$ is multiplied with $\sqrt{\Delta t}$ and a learnable parameter $\sigma^{\boldsymbol \phi}$ to realize matrix elements $\sim \mathcal N(0,(\sigma^\boldsymbol \phi)^2 \Delta t)$. We will add this discussion to the appendix.
>
> > Minor. You followed [47] (in the paper) or [48] (in the appendix) for preprocessing the human activity dataset. Which one did you follow?
>
> Thank you for pointing this out. We followed [47] and will update the manuscript accordingly.

---

> > ### Comment · Reviewer_GYzQ · 2023-08-18
> >
> > Thank you very much for your thorough response and for addressing all my concerns.

---

### Author Rebuttal · Authors · 2023-08-08

We like to thank **all** reviewers for their overall positive feedback, valuable comments and suggestions!

While we address all issues point by point per reviewer, we first like to clarify issues that are common to (almost) all reviews: (&#9824;) to substantiate the computational claims from the manuscript with a **runtime comparison**, (&#9829;) to resolve shortcomings in the **presentation of our preliminaries** and (&#9830;) to clarify the **optimization objective** for per time-point classification/regression experiments.

---
*We refer to the following works in our rebuttal and will denote figures and tables from the attached PDF as R-Fig. & R-Tbl. XXX, respectively.*

**[Rubanova et al., 2019]** Y. Rubanova, R.T.Q. Chen, and D. Duvenaud.
*Latent ODE for irregularly-sampled time series*. In: NeurIPS 2019.

**[Xi et al., 2020]** X. Li, T.-K. L. Wong, R.T.Q. Chen, and D. Duvenaud. *Scalable gradients for stochastic differential equations*. In: AISTATS 2020.

**[Schirmer et al., 2022]** M. Schirmer, M. Eltayeb, S. Lessmann, and M.  Rudolph. *Modeling irregular time series with continuous recurrent units*. In: ICML 2022.

**[Marjanovic et al., 2018]** G. Marjanovic and V. Solo. *Numerical methods for stochastic differential equations in matrix Lie groups made simple*. IEEE Trans. Autom. Control., 63(12):4035–4050, 2018.

**[Muniz et al., 2022]** M. Muniz, M. Ehrhardt, M. Günther, and R. Winkler. *Higher strong order methods for linear Itô SDEs on matrix lie groups. BIT Numer. Math., 62(4):1095–1119, 2022.*

**[Itô, 1975]** Kiyosi Itô. Stochastic Calculus. Lect. Notes Phys., 39:218–223, 1975.

---

(&#9824;) **Runtime comparison.** To back up our computational claims, we ran additional experiments to assess the runtime and efficiency of our method, summarized in the table below (and **R-Tbl. 1**).

| | Runtime/Batch [s] | Test MSE $\left(\times 10^{-3}\right)$ |
| ---------------------------- | ----------------- | -------- |
| LODE (Rubanova et al., 2019) | 0.053 $\pm$ 0.004 | 14.9 $\pm$ 0.275 |
| LSDE (Xi et al., 2020)       | 0.112 $\pm$ 0.009 | 14.0 $\pm$ 0.543 |
| Ours - $(\mathbb{S}^{n-1},\mathrm{SO}(n))$ | 0.055 $\pm$ 0.005 | 11.2 $\pm$ 0.573 |
| | | |
| Ours - $(\mathbb{R}^{n},\mathrm{GL}(n))$ | 0.056 $\pm$ 0.008 | 12.9 $\pm$ 0.854 |

This comparison is done on _Rotating MNIST_, using the same architecture (encoder/decoder) as in Sec. 4.2 of the manuscript, but the latent dynamic models vary. We compare our approach to 1) a latent ODE (_LODE_, as in [Rubanova et al., 2019] and 2) a latent SDE (_LSDE_, as in [Xi et al., 2020]. All models have approx. 450k parameters. Runtime/Batch refers to the wall clock time per **forward+backward pass** (with batch size 50), computed from a list of all SGD update steps from 5 randomly initialized training runs. We also list the final test MSEs, averaged over these 5 runs. For a fair comparison, we always use *Euler's method* as ODE/SDE solver with a fixed step size of 1/16 (since 16 time points are available in [0,1]).

Overall, although our approach implements a latent **SDE**, runtime per batch is on par with the latent ODE variant and substantially lower than the more flexible latent SDE approach from [Xi et al., 2020] (which is closest in terms of modeling choice). Notably, decreasing the step size (e.g., to 1e-2) for latent ODE/SDE did not noticeably improve performance (MSE) but increased runtime linearly.

To account for potentially different convergence speeds, we also present loss curves in **R-Fig. 3**. The left plot in this figure shows *training MSE *vs.* epoch* and reveals that our method needs fewer SGD updates to converge, whereas LSDE and LODE converge at approx. the same rate. The right plot accounts for different runtimes per batch and reveals that our method needs only around half of the training time as LSDE.

*Similar runtime measurements for all experiments will be added to a final version.*

(&#9829;) **Presentation of preliminaries.** We agree that our current presentation of the preliminaries in Sec. 3.1 could be misunderstood, especially with the introduction of $h$ in Eq. (1). The arguably more common approach (which we will follow) is to state the data generation process as (1) sampling a realization of a path-valued latent variable $\mathbf{z}$ from a suitable parametric prior $p_{\boldsymbol{\theta}^*}(\mathbf{z})$ and subsequently (2) sampling a realization of a path-valued observation $\mathbf{x}$ from some conditional distribution $p_{\boldsymbol{\theta}^*}(\mathbf{x}|\mathbf{z})$. This makes the use of $h$ obsolete. We apologize for the confusion and will adjust Sec. 3.1 accordingly.

(&#9830;) **Clarification of optimization objective(s).**
We first note that our data (irrespective of the task) always contains multiple series of time-indexed observations and that during training, we minimize the negative ELBO, consisting of a KL divergence term between posterior and prior latent path distributions and a log-likelihood term $\log p_{\boldsymbol{\theta}}(\mathbf{x}|\mathbf{z})$. If additional supervision is available, a corresponding loss term is added, e.g., (per time-point) cross-entropy for classification or (per time-point) MSE for regression. These losses are computed on the output of an additional neural network (typically a two-layer MLP) that maps the **latent states** $\mathbf{z}$ (*not* the reconstructions) to the response variable(s). In practice, when evaluating $\log p_{\boldsymbol{\theta}}(\mathbf{x}|\mathbf{z})$, we model the conditional distribution at each time point as a Gaussian distribution. However, this is just one of many possible choices.

The only exception to the statement above is the **Human Activity** experiment, where we follow [Rubanova et al., 2019] for fair comparison and train solely with cross-entropy and the KL term, but without $\log p_{\boldsymbol{\theta}}(\mathbf{x}|\mathbf{z})$.

---

### Decision · Program_Chairs · 2023-09-21

**Decision:**

Accept (poster)

**Comment:**

The reviewers agree that this would make an interesting contribution, with the author responses having generally strengthened the reviewers' opinions of the paper. Please read the reviews carefully, as well as your responses, and make the necessary changes in the camera-ready version.